# A release-and-capture mechanism generates an essential non-centrosomal microtubule array during tube budding

Ghislain Gillard [1], Gemma Girdler[1,2] & Katja Röper [1]✉

Non-centrosomal microtubule arrays serve crucial functions in cells, yet the mechanisms of their generation are poorly understood. During budding of the epithelial tubes of the salivary glands in the *Drosophila* embryo, we previously demonstrated that the activity of pulsatile apical-medial actomyosin depends on a longitudinal non-centrosomal microtubule array. Here we uncover that the exit from the last embryonic division cycle of the epidermal cells of the salivary gland placode leads to one centrosome in the cells losing all microtubule-nucleation capacity. This restriction of nucleation activity to the second, Centrobin-enriched, centrosome is key for proper morphogenesis. Furthermore, the microtubule-severing protein Katanin and the minus-end-binding protein Patronin accumulate in an apical-medial position only in placodal cells. Loss of either in the placode prevents formation of the longitudinal microtubule array and leads to loss of apical-medial actomyosin and impaired apical constriction. We thus propose a mechanism whereby Katanin-severing at the single active centrosome releases microtubule minus-ends that are then anchored by apical-medial Patronin to promote formation of the longitudinal microtubule array crucial for apical constriction and tube formation.

[1] MRC-Laboratory of Molecular Biology, Cambridge, UK. [2] Present address: Wellcome-MRC Stem Cell Institute, Jeffrey Cheah Biomedical Centre, University of Cambridge, Cambridge, UK. ✉email: kroeper@mrc-lmb.cam.ac.uk

The microtubule cytoskeleton plays many essential roles in cells, from faithful chromosome segregation during cell division to the transport of many cargoes. In most animal cells that are actively dividing and cycling, the microtubule cytoskeleton is nucleated and anchored at centrosomes throughout interphase but especially during mitosis[1]. Centrosomes consist of a single or a pair of centrioles at their core, depending on the stage of the cell cycle, surrounded by a cloud of pericentriolar material (PCM) that contains the critical microtubule nucleator γ-tubulin in form of the γ-tubulin ring complex (γ-TURC) that templates the microtubule protofilament arrangement[2].

However, in post-mitotic cells such as neurons and epithelial cells, microtubules can also be nucleated or anchored from non-centrosomal sites. Non-centrosomal microtubule function in those cells is crucial for processes such as directed intracellular transport, organelle positioning, and cell polarity[3–5]. In post-mitotic epithelial cells, non-centrosomal microtubules can be organised in different arrays, lying for instance parallel to the apical surface[6,7] or forming extended longitudinal arrays along the apical–basal axis[8,9]. There is now growing evidence for a role of microtubules in epithelial morphogenesis. They can do so by exerting forces against the plasma membrane[10], by coordinating forces at the tissue scale[6,7], or by regulating actomyosin localisation or activity[8,11]. Despite such important cellular and developmental functions of non-centrosomal microtubules, it remains unclear, though, what the mechanism of non-centrosomal microtubule generation is, whether it involves for instance nucleation from non-centrosomal MTOCs, or whether pre-existing microtubules become relocalised[12].

We have previously shown a function for a longitudinal non-centrosomal microtubule array during tube morphogenesis in the *Drosophila* embryo[8]. During the initial tissue bending and budding process of the tubes of the salivary gland from an epithelial placode (Fig. 1a–a‴), microtubules rearrange by 90° from a previously apical centrosomal to a longitudinal non-centrosomal array[8] (Supplementary Fig. 1). This happens concomitantly with the cells undergoing apical constriction (Fig. 1a″), which itself is required to drive the budding morphogenesis[8,13]. Disruption of the microtubule cytoskeleton leads to the selective loss of an apical-medial, but not junctional, pool of actomyosin, and this pool is required for successful apical constriction[8]. It is thus far unclear what triggers and establishes the microtubule rearrangement and generation of the non-centrosomal array. Disruption of the wild-type pattern of apical constriction early on in the placode, e.g. when microtubules are depleted, leads to aberrant gland and lumen shapes at later stages when all secretory cells have internalised[8,13].

*Drosophila* embryos undergo a modified fast cell cycle in early embryogenesis[14]. At the beginning of tissue morphogenesis, divisions become asynchronous and most cells in the embryo only divide three more times[15]. This leads the epidermal cells after each M-phase in these cycles to inherit two separated centrosomes consisting of a mother centriole with an attached pro-centriole or daughter[16]. At the time point that salivary gland morphogenesis commences in the embryo, the salivary gland placodal cells differ from all other epidermal cells in that they are the first to finish all embryonic division cycles and are the first cells to enter a G1 phase in interphase of embryonic cell cycle 17 and concomitantly become post-mitotic[15]. At the end of the last mitosis 16, in contrast to the previous cycles, these cells now inherit two centrosomes consisting of an isolated centriole each, with no pro-centriole or daughter[16].

Here, we describe our discovery of a step-wise process that implements these changes in the salivary gland placode: as part of concluding embryonic mitoses, the cells of the placode are the first to enter a G1 phase with concomitant loss of microtubule nucleation capacity of the centrosome exhibiting low levels of Centrobin, a loss that we show is important to ensure proper morphogenesis. Furthermore, our results suggest that microtubules generated from the remaining active centrosome are released by severing through Katanin and then anchored and stabilised by Patronin, the *Drosophila* CAMSAP homologue[9,17,18], that is quickly recruited to free microtubule minus-ends in placodal cells. Both Katanin and Patronin function, we show, is required for microtubule rearrangements, for the proper apical-medial actomyosin activity and apical constriction during early tube budding morphogenesis.

## Results

**Changes to centrosomal microtubule-nucleation capacity in the placode.** In order to investigate whether changes at centrosomes contributed to the formation of the non-centrosomal microtubule array in the salivary gland placode, we decided to analyse levels of centrosome components. The overall capacity and requirement for centrosomal versus non-centrosomal microtubule nucleation during *Drosophila* embryogenesis, beyond the fast synchronised cell cycles 1–13 at very early stages, is unclear. Maternal loss of key centrosome components leads to developmental arrest of early embryos during the fast syncytial divisions[19], whereas in zygotic mutants the protein levels of centrosome components only run out in larval stages and embryogenesis is unaffected[20]. Upon specification, the cells of the salivary gland primordium, the placode (Fig. 1a, a″), have completed mitosis 16 and cease full mitoses for the remainder of embryogenesis. They enter their first G1 phase during nuclear cycle 17 and remain in it until the end of embryogenesis, hence being post-mitotic[16]. Amongst cells of the anterior ectoderm, the salivary gland placodal cells are in fact the first organ primordium to reach this stage (Fig. 1b–b″). Concomitantly, salivary gland placodal cells now enter endoreduplication or endocycles, i.e. DNA-synthesis and segregation without any accompanying cytokinesis, leading to most salivary gland cells being polytene at larval stages[21].

Interestingly, labelling of centrosomes within the salivary gland placode at late stage 10, just prior to starting off the budding morphogenesis, revealed that most post-mitotic placodal cells contained two well-separated centrosomes that showed a striking asymmetry in the accumulation of Centrosomin (Cnn), a component of the PCM key to centrosome maturation (Fig. 1b–e; 45.7% of cells with an asymmetry of Cnn accumulation and another 42.9% with Cnn completely restricted to one centrosome)[22,23]. Most cells in the surrounding epidermis at this stage have not completed cell divisions, reflected by the continuing occurrence of mitoses and accompanying clusters of high mitotic Cnn labelling (Fig. 1b″, f″ arrows, and g, g′), as the microtubule nucleation capacity of centrosomes increases dramatically during mitosis, concomitant with a drastic increase in PCM[24]. Actively dividing cells just after M-phase will also show Cnn asymmetry as the daughter centrosome upon division needs to re-recruit PCM over the following S-phase[25]. Hence, many cycling epidermal cells also displayed a temporary Cnn asymmetry (Fig. 1d, e; analysed in parasegment 3 (PS3): 66.2% of cells with asymmetric Cnn accumulation and 22.1% of cells with Cnn restricted to one centrosome). Such epidermal asymmetry was evident from stage 9 onwards (Supplementary Fig. 2), representing cells in G2 of cycle 15 or 16 that are in the process of PCM recruitment about to generate two centrosomes with equal Cnn prior to the next M-phase. Accordingly, at mid-stage 11, half of the epidermal cells in parasegment 3 had entered mitosis again (48.6%) and 20.0% in addition showed equal Cnn

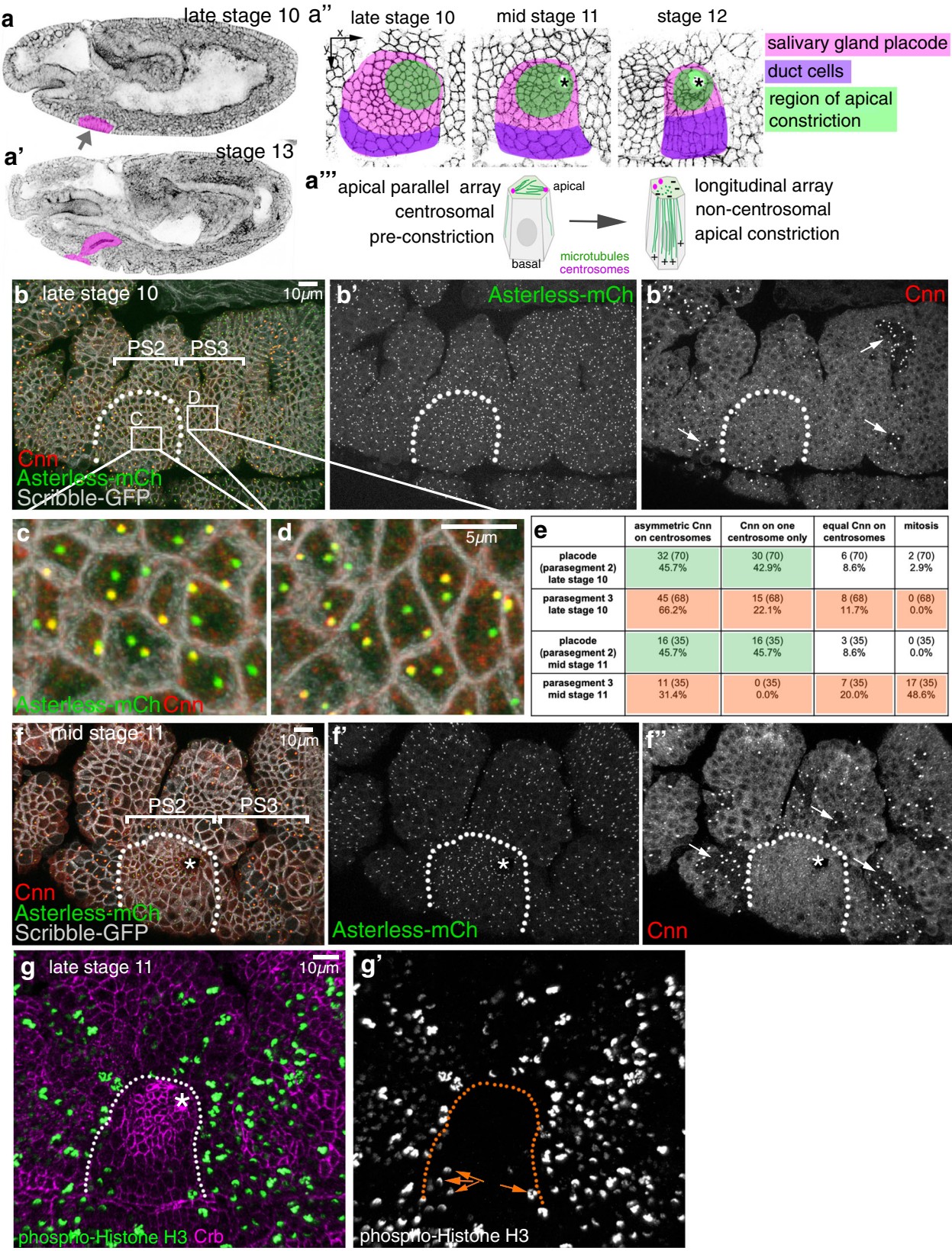

distribution (Fig. 1e, f″), thus most likely being in G2 just prior to the next mitosis. In contrast, in the already post-mitotic placodal cells the Cnn asymmetry remained identical (Fig. 1e). Note that cells close to the ventral midline, that will later form part of the duct of the salivary gland (Fig. 1a″), re-entered mitosis once more, as evident by phospho-histone H3 labelling at late stage 11

(Fig. 1g, g′). In contrast to Cnn, Asterless (Asl), a core component of the centriole, was equally enriched on both centrosomes throughout the epidermis and placode (Fig. 1b–f′).

We also investigated the distribution of other centrosome components in placodal cells and found that the core centriole components Asl, Spd-2 and Sas-4 (Fig. 2a) were equally enriched

**Fig. 1 Changes to centrosomes during tube budding. a–a′′′** The salivary glands form from two epithelial placodes localised on the ventral surface of the *Drosophila* embryo that become specified at the end of embryonic stage 10 (**a**). These placodes invaginate through budding to form a simple tube (**a′**). **a′′** During invagination, cells close to the forming invagination point (asterisks) in the dorsal-posterior corner constrict apically (green cells) as part of the morphogenetic programme[13]. Pink marks all secretory cells of the placode, magenta marks future duct cells near the ventral midline. **a′′′** Concomitant with constriction, placodal cells undergo a rearrangement of their microtubule cytoskeleton, from centrosomally anchored bundles running parallel to the apical surface to a non-centrosomal longitudinal array with minus ends apically. **b–f′′** Salivary gland placodal cells in embryonic parasegment 2 (**b**, **f**) are the first epithelial tissue primordium to enter G$_1$ phase of embryonic division cycle 17. This post-mitotic state leads to a permanent loss of Cnn (**b′′**, **f′′**) from one centrosome after the last mitosis (quantification in **e**). Arrows in **b′′** and **f′′** point to mitotic domains in the epidermal tissue surrounding the placode. **c** and **d** illustrate the permanent (PS2) and temporary (PS3) Cnn asymmetry in cells; note that epidermal cells in the domain shown in **d** have entered mitosis again in **f′′**. **g–g′** Phospho-histone H3 labelling (green and single channel in **g′**) to mark mitotic cells at late stage 11 clearly shows that the secretory cells of the salivary gland placode have stopped dividing, only a few duct cell precursors near the ventral midline still undergo mitoses (orange arrows in **g′**), whereas epidermal cells in the surrounding epidermis are still undergoing division cycle 16. The salivary gland placode is indicated by dotted lines and the invagination point, were present, by an asterisk. See also Supplementary Figs. 1 and 2.

on both centrosomes (Fig. 2b, c′). However, in addition to the asymmetric distribution of Cnn, γ-tubulin, that is recruited by Cnn[26], and Polo-kinase, a key regulator of PCM recruitment to the centrosome[22,27], were enriched asymmetrically (Fig. 2b, c′ and Supplementary Fig. 3). This asymmetry explained the lack of nucleation capacity of some of the centrosomes analysed in live assays using EB1-GFP to label growing microtubule plus-ends and Asl-mCherry (Supplementary Fig. 3a–c). Consistent with the asymmetry in nucleation capacity, 90% of centrosomes with higher levels of Cnn within the placode actively nucleated microtubules in live assays (Fig. 2d, e), underlining that only Cnn-enriched centrosomes still retain nucleation capacity. Therefore, the post-mitotic placodal cells displayed a pronounced and permanent centrosome asymmetry and hence asymmetry in the capacity to nucleate microtubules.

How is this interphase asymmetry in centrosomes achieved? Similar to the salivary gland placode, in *Drosophila* larval neuroblast only one centrosome (composed at this stage of a single centriole as in the salivary gland placode) nucleates an interphase array, and this is the daughter centrosome/centriole[28]. This centriole is selectively marked by the daughter-specific centriole component Centrobin[29]. In neuroblasts, upon finishing a mitosis, the mother centriole/centrosome initially contains more PCM including Cnn, but this is then lost over time whilst the daughter centriole/centrosome that contains Centrobin continues to accumulate more and more PCM[30]. To test whether a similar mechanism could be at work in the placode, we analysed the distribution of Centrobin, using *Ubi::Centrobin-YFP*[29], in the placodal cells. Centrobin was highly asymmetric (Fig. 2c, c′) and was always enriched on the one centrosome in placodal cells that also contained higher levels of Cnn (Fig. 2f–f′′′′), as visualised in line scans through centrosomes of a single cell (Fig. 2g, Supplementary Fig. 3d–d′). Polo-kinase, γ-tubulin and Cnn were as expected enriched on the same centrosome in placodal cells, though the Polo and γ-tubulin asymmetry were not as pronounced as the difference in Cnn (Fig. 2c′, Supplementary Fig. 3e–j).

Taken together these results show that only a single centrosome in the secretory cells of the salivary gland placode during early tube morphogenesis retains microtubule nucleation capacity. Similar to fly neuroblasts, Centrobin-labelling suggests that this could be the daughter centriole.

**Centrosome asymmetry is a prerequisite for proper morphogenesis.** We next sought to determine whether centrosome asymmetry and restriction to nucleation from a single centrosome were important for the generation of the non-centrosomal array or the morphogenesis. To address this question, we analysed embryos overexpressing a transgene of γ-tubulin expressed under the *ncd* promoter that is expressed at high levels up to mid-

embryogenesis (*ncd::γ-tubulin-EGFP*; Fig. 3a, b). In such embryos the γ-tubulin centrosomal asymmetry in placodal cells was less pronounced (Fig. 3a′, b), suggesting that the overexpression led to a more equal γ-tubulin accumulation at both centrosomes. γ-tubulin-EGFP was still restricted to centrosomes only. Interestingly, the whole embryonic epidermis where *ncd::γ-tubulin-EGFP* was expressed showed an increase in microtubule intensity compared to controls. This was particularly pronounced in the salivary gland placodal cells overexpressing γ-tubulin-EGFP, as measured through labelling of either α-tubulin (Fig. 3c and Supplementary Fig. 4a, b′) or tyrosinated α-tubulin labelling (Fig. 3c, d′). Labelling for stable microtubules (using acetylated α-tubulin staining) did not increase but was rather slightly reduced (Fig. 3c, e, e′), suggesting the overall increase reflected newly polymerised microtubules.

As the future secretory cells of the placode are undergoing apical constriction at this time point, we also analysed actomyosin levels in embryos, as actomyosin drives the apical constriction and depends on microtubules in these cells[8]. In addition to an increase in overall microtubule intensity, levels of F-actin also increased strongly, and this was again particularly enhanced in the secretory cells for both junctional and apical-medial actin (Fig. 3f, f′ and Supplementary Fig. 4e). Furthermore, cell apices in *ncd::γ-tubulin-EGFP* placodes often showed a convoluted apical junctional morphology ('wavy' junctions; Fig. 4a–e), that is indicative of strong, possibly excessive, apical constriction[31,32]. In line with this finding, cells showed more constricted apices compared to control (Fig. 4), also reflected by an increase in myosin II levels (revealed by an endogenously tagged version of myosin regulatory light chain, Sqh-RFP[33]; Fig. 4g and Supplementary Fig. 4c, d′).

Excessive amounts of microtubules generated in placodal cells of *ncd::γ-tubulin-EGFP* embryos appeared to contribute to an enlarged non-centrosomal array (Fig. 4h, i), also visible in cross-sections of placodes (Supplementary Fig. 4f, g′). This was confirmed by an increased accumulation of apical-medial compared to junctional Patronin-RFP, with Patronin being a bona fide minus-end binding protein (Fig. 4j, l and Supplementary Fig. 4h, i′′′[34,35] and see below). This excess amount of microtubules seemed to be the consequence of an increased proportion of centrosomes observed nucleating microtubules in γ-tubulin-EGFP-overexpressing embryos (Supplementary Fig. 4j–ll′′ and Supplementary Movie 2). We suggest that this increased non-centrosomal array is the underlying cause of the increase in apical-medial actomyosin and increased apical constriction (Fig. 4m).

Thus, not only did post-mitotic placodal cells show a distinctive centrosome asymmetry, similar to that previously only observed in neuroblasts, but this asymmetry appeared important for the establishment of a non-centrosomal microtubule array with the

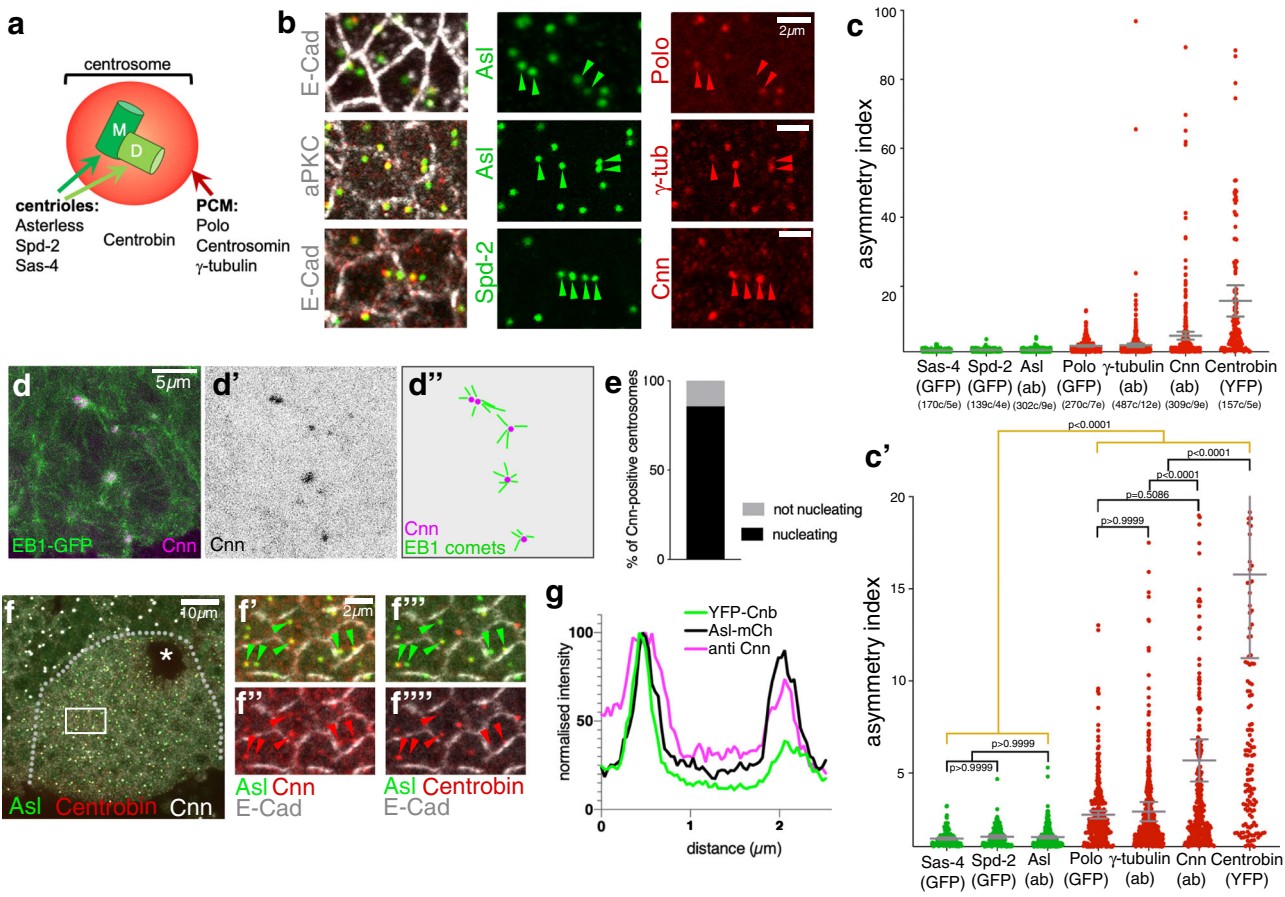

**Fig. 2 Placodal centrosomes show strong asymmetry of PCM constituents and microtubule nucleation capacity. a** Centrosomes are usually built of two centrioles, a mother centriole (M) inherited from the last division, and a newly nucleated daughter centriole (D), surrounded by a cloud of PCM. Key centriole components: Asl, Spd-2, Sas-4, Centrobin (only on daughter centriole); PCM components: Polo-kinase, Cnn and γ-tubulin. **b, c'** Whereas the centriole components Asl, Spd-2 and Sas-4 are equally enriched on both centrosomes in placodal cells, PCM components Polo, Cnn and γ-tubulin show asymmetric accumulation at placodal centrosomes (localisation for all in **b** and quantification in **c, c'**). Cell outlines are marked by E-Cadherin (E-Cad) or aPKC in **b. c, c'** show the same data, with **c'** showing a zoomed version of the 0–20 values of the asymmetry index of **c**. *n* values are: *Sas4-GFP*: 170 cells from 5 embryos; *Spd-2-GFP*: 139 cells from 4 embryos; Asl (antibody, ab): 302 cells from 9 embryos; *GFP-Polo*: 270 cells from 7 embryos; γ-tubulin (ab): 487 cells from 12 embryos; Cnn (ab): 309 cells from 9 embryos; *Cnb-YFP*: 157 cells from 5 embryos. Pairwise comparison of distributions was done via Kruskal–Wallis test (one-way ANOVA, all *p* values of these are listed in the "Methods" section). Mean and 95% CI are shown. Here and in all following quantifications cell (**c**) and embryo (**e**) numbers are indicated below the plots. **d, e** 90% of Cnn-containing centrosomes actively nucleate microtubules. **d** Projection of 30 consecutive time frames, 0.55 s apart, of a time lapse movie of *EB1-GFP Cnn-RFP* flies. **d'** shows the Cnn-RFP channel to indicate the positions of centrosomes. **d"** schematically illustrates the EB1 comets moving away from Cnn-positive centrosomes that indicate active microtubule nucleation. **e** Quantification of nucleation capacity of 70 Cnn-positive centrosomes. See Supplementary Movie 1. **f–g** Cnn and Centrobin (Cnb) accumulate asymmetrically in individual cells, but on the same centrosome. Box in **f** is shown enlarged in **f'–f"**. Asl labels all centrosomes and E-Cad labels cell outlines in **f'–f""**. **g** Line scan profile through both centrosomes of a single cell illustrates the co-enrichment of Cnb and Cnn on the same centrosomes. More line scan examples are shown in Supplementary Fig. 3. The salivary gland placode is indicated by a dotted line and the invagination point by an asterisk in **f**. Filled arrowheads in **b** and **f'–f""** indicate centrosomes with PCM and Centrobin accumulation, respectively, while hollow arrowheads indicate centrosomes without PCM or Centrobin staining. See also Supplementary Fig. 3.

right amount of microtubules. Our data suggest that an excessive microtubule array, by enhancing constriction across the placode and interfering with the wild-type pattern of it, can interfere with wild-type tube budding (see below).

**Katanin accumulates at the apical-medial site of placodal cells.** Formation of non-centrosomal microtubule arrays in epithelial cells has been proposed to depend on the relocalisation of nucleation capacity away from centrosomes, with these centro-somes being 'switched off'[12,36]. However, the retention of microtubule-nucleation capacity at one centrosome per placodal cell suggested an alternative mechanism could be at hand. Fur-thermore, it has been proposed that non-centrosomal micro-tubules could be generated via the release of centrosomal

microtubules by severing enzymes followed by selective capture and stabilisation of microtubule minus-ends[36,37]. One such severing enzyme is Katanin, which is a multi-subunit microtubule severing enzyme[38]. Using a YFP-exon trap line of the regulatory subunit Katanin 80, a line tagging the endogenous locus and thus faithfully reporting expression and localisation of the endogenous protein, we discovered that Katanin 80 was selectively enriched in bright foci in the placodal cells compared to the surrounding epidermis from early stage 11 onwards (Fig. 5a–a"). Within the placodal cells about to or undergoing apical constriction, foci of Katanin 80-YFP were localised in an apical-medial position (Fig. 5a"). This medial localisation was close to centrosomes in just under half of all cells that showed a clear Katanin 80-YFP signal, and the localisation was more diffuse or junctional in other

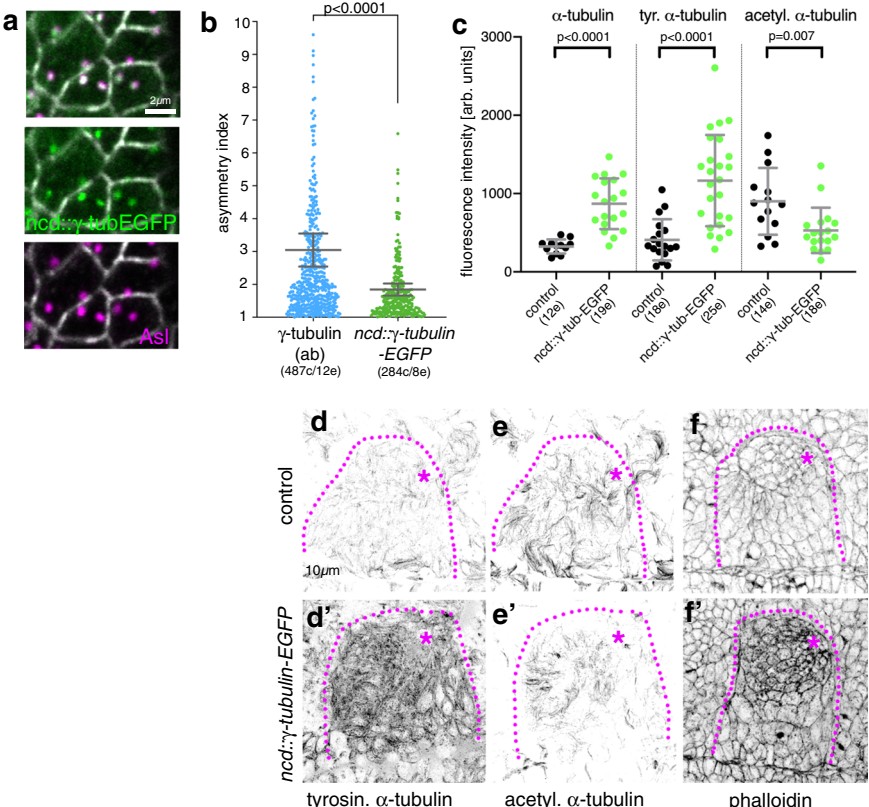

**Fig. 3 γ-tubulin overexpression increases microtubule and actomyosin levels. a, b** Overexpression of γ-tubulin using *ncd::γ-tubulin-EGFP* leads to reduction of γ-tubulin asymmetry at centrosomes (localisation in **a–a″** and quantification in **b**). **b** *ncd::γ-tubulin-EGFP* asymmetry was quantified as part of the data illustrated in Fig. 2c, c′. The γ-tubulin (ab) data are reproduced here in comparison to the *ncd::γ-tubulin-EGFP* overexpression: with *n* = 284 cells from 8 embryos; statistical significance was deduced by two-sided Mann–Whitney test of comparison as *p* < 0.0001, shown are mean and 95% CI. **c** Overexpression of *ncd::γ-tubulin-EGFP* and loss of γ-tubulin asymmetry leads to increased levels of microtubule labelling using antibodies against α-tubulin or tyrosinated α-tubulin, but a slight reduction in acetylated α-tubulin labelling. Analysed were: 12 embryos for control and 19 embryos for *ncd::γ-tubulin-EGFP* for α-tubulin, 18 embryos for control and 25 embryos for *ncd::γ-tubulin-EGFP* for tyrosinated α-tubulin; 14 embryos for control and 18 embryos for *ncd::γ-tubulin-EGFP* for acetylated α-tubulin. Statistical significance was determined by two-sided unpaired *t*-tests with Welch's correction for α-tubulin (*p* < 0.0001) and tyrosinated α-tubulin (*p* < 0.0001) and a two-sided Mann–Whitney test for acetylated α-tubulin (*p* = 0.007), shown are mean ± SD. **d–f′** Labelling of stage 11 placodes of control and γ-tubulin overexpressing embryos, showing the increase in tyrosinated α-tubulin (**d, d′**) and phalloidin labelling to reveal F-actin (**f, f′**) and the decrease in acetylated α-tubulin labelling (**e, e′**). See also Supplementary Fig. 4.

cells (Fig. 5b–b′). Furthermore, Katanin 80-YFP near centrosomes colocalised with the apical foci demarcating ends of microtubules bundles (Fig. 5c). In cells where Katanin localised near a single centrosome, this was the one enriched in Cnn and thus the one still nucleating microtubules (Fig. 5d, d′). When we imaged microtubules near centrosomes live in placodal cells using *Jupiter-GFP Asl-mCherry* flies and imaging in a single apical focal plane comprising the centrosome position, we frequently observed bright foci of Jupiter-GFP moving away from the Asl-mCherry-labelled centrosome (Fig. 5e, f). Such movement within a single plane was consistent with newly released microtubule minus-ends moving away from centrosomes.

Thus, Katanin's expression and localisation in placodal cells strongly suggested that it plays a role in the generation of the non-centrosomal microtubule array, possibly via severing and release of microtubules from the nucleating centrosome.

**Loss of Katanin disrupts the non-centrosomal microtubule array and apical-medial actomyosin-mediated apical constriction.** In order to test whether Katanin-severing was important for the generation of the non-centrosomal microtubule array within the placodal cells and for tube budding morphogenesis, we selectively depleted Katanin 80-YFP from the placodal cells using the

degradFP system (Fig. 6a). This system can target GFP- and YFP-tagged endogenous proteins for degradation by the proteasome through tissue-specific expression of a modified F-box protein that is fused to an anti-GFP nanobody[39] (Fig. 6a). Expression of degradFP specifically in the salivary gland placode under *fkhGal4* control led to a reduction of Katanin80-YFP levels in the placodal cells (Fig. 6b–d). In contrast to the control (Fig. 6e, e′), in these Katanin-depleted cells microtubules did not lose contact with centrosomes but rather remained in close association with the centrosomes labelled by Asl (Fig. 6f, f′). This led to microtubule bundles running parallel to the apical domain, rather than the foci of microtubule minus ends being visible apically as in the control (Fig. 6e–g).

We have previously shown that overall loss of microtubules in the placodal cells led to a failure of apical constriction of the cells during budding, due to a selective loss of apical-medial actomyosin[8]. We therefore analysed whether failure in the generation of the longitudinal non-centrosomal array and continued centrosomal anchoring as seen under Katanin-depletion affected apical constriction. We quantified the apical cell area of placodal cells at a timepoint where the invagination pit had formed (mid-to-late stage 11; Fig. 6h–h‴). Though apical constriction was not abolished, there was a significant reduction in the amount of apically constricted cells when Katanin was

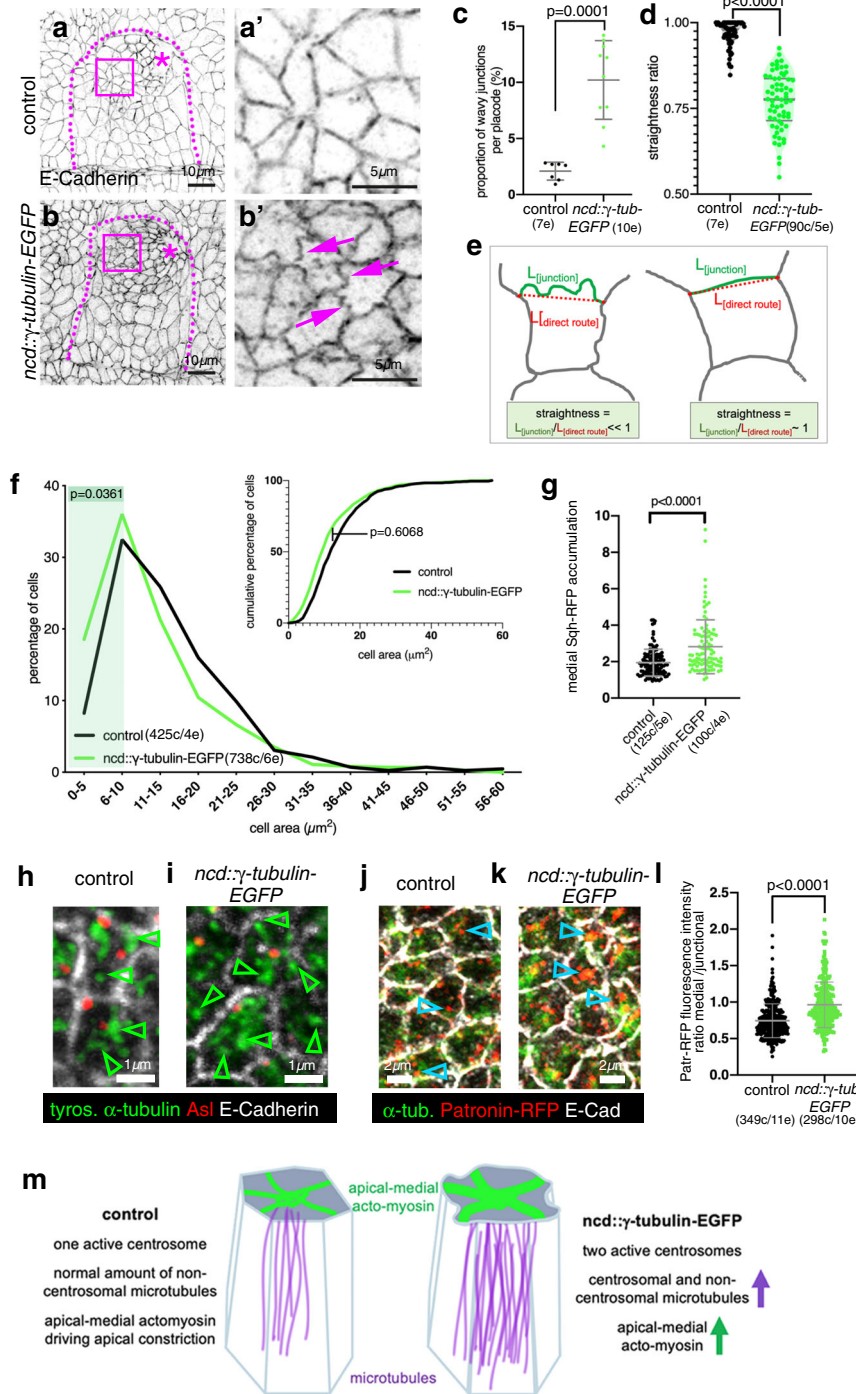

degraded (Fig. 6h″–h‴). Furthermore, analysis of apical actin accumulation revealed that apical-medial actin levels were reduced (Fig. 6i–k).

Thus, Katanin-mediated microtubule severing at a single active centrosome appears to plays a key role in the formation of the longitudinal non-centrosomal microtubule array that in turn supports apical constriction.

**Patronin localises specifically to non-centrosomal microtubule minus-ends in apical-medial region of placodal cells.** Free microtubule minus-ends generated by Katanin severing activity tend to be highly susceptible to depolymerisation and usually require stabilisation to prevent this happening[40,41]. Members

of the family of CAMSAPs (Calmodulin-regulated-Spectrin-associated proteins), with the single orthologue in flies being Patronin, have such a capacity for minus-end stabilisation and also anchoring at non-centrosomal sites[34,35]. Endogenously tagged Patronin (Patronin-YFP)[18] accumulated near adherens junctions in most epidermal cells outside the placode (Fig. 7a–c′), consistent with a previously described function in stabilising microtubules in this location[42]. Within the placodal cells undergoing apical constriction, though, Patronin accumulated in an apical-medial position in a dynamic way (Fig. 7a, b′ and Supplementary Fig. 5). Indeed, medial Patronin foci in cells near the invagination point were dynamic and coalesced during an apical constriction pulse (Supplementary Fig. 5d–f), reminiscent of the pulsatile behaviour of apical-medial actomyosin that drives the

**Fig. 4 γ-tubulin overexpression leads to excessive apical constriction in the placode. a, b′** E-Cadherin-labelled placodes of control and γ-tubulin-overexpressing embryos. Magenta boxed areas in **a, b** are magnified in **a′, b′**. Dotted lines denote placode boundary, asterisks the invagination pit. Magenta arrows in **b′** point to wavy junctions in ncd::γ-tubulin-EGFP embryos. **c–e** Quantification of junction waviness. **c** Placodes of ncd::γ-tubulin-EGFP embryos show a significantly higher proportion of wavy junctions than control placodes (7 placodes for control and 10 placodes for ncd::γ-tubulin-EGFP were analysed); $p = 0.0001$, determined by two-sided unpaired Mann–Whitney t-test, shown are mean ± SD. **d** Wavy junctions in ncd::γ-tubulin-EGFP overexpressing embryos are significantly less straight (90 junctions from 5 placodes) than randomly picked junctions in control placodes (90 junctions from 5 placodes). Statistical significance was determined using two-sided unpaired Mann–Whitney t-test with $p < 0.0001$, shown are violin plots with median and quartiles. **e** The straightness of a junction is defined as the ratio of the length of the junction itself ($L_{[junction]}$) divided by the length of the direct route between vertices ($L_{[direct\ route]}$)[31,70]. For a straight junction this value is close to 1, for a wavy junction it is «1. **f** Apices of secretory cells of ncd::γ-tubulin-EGFP embryos are more constricted than apices of control placodes, illustrated are both percentage of cells of a certain apical area bin as well as the cumulative percentage of cells. 738 cells were analysed in 6 ncd::γ-tubulin-EGFP embryos and 425 cells in 4 control placodes; Kolmogorov–Smirnov-two-sample test on the cumulative data did not show a significant difference between control and γ-tubulin overexpressing embryos ($p = 0.6068$). However, comparing the distribution of cells with small apical areas between 0 and 5 $\mu m^2$ ($p = 0.0361$) and cells with apical areas in a range between 5 and 10 $\mu m^2$ ($p = 0.0361$), showed a significant difference, indicated by green shaded area. **g** ncd::γ-tubulin-EGFP embryos show increased apical-medial myosin compared to control (visualised using sqh-RFP). 100 cells in 4 placodes were analysed in ncd::γ-tubulin-EGFP embryos and 125 cells in five control placodes; statistical significance was deduced by two-sided unpaired Mann-Whitney test as $p < 0.0001$, shown is mean ± SD. **h, i** Microtubules within the apical-medial region of placodal cells of ncd::γ-tubulin-EGFP embryos remain organised in a non-centrosomal fashion (**i**), as in the control (**h**). Arrowheads point to ends of microtubules or microtubule bundles away from centrosomes. **j–l** An increased amount of apica-medial Patronin-RFP, a stabiliser of free microtubule minus-ends, accumulates in placodal cells of ncd::γ-tubulin-EGFP embryos where increased numbers of microtubule bundle foci are found (**k**) in comparison to control (**j**), indicated by blue arrowheads. **l** Quantification of medial to junctional Patronin-RFP intensity; 298 cells in 10 placodes were analysed in ncd::γ-tubulin-EGFP embryos and 349 cells in 11 control placodes; statistical significance was deduced by two-sided unpaired Mann–Whitney test as $p < 0.0001$, shown is mean ± SD. **m** Model of the effect of γ-tubulin overexpression in placodal cells. See also Supplementary Fig. 4.

apical constriction[8]. This suggests a dynamic behaviour of a hub of proteins involved in the constriction of the apical-medial domain of placodal cells.

Just upon salivary gland placode specification at the end of stage 10, but prior to the microtubule rearrangement, Patronin was also localised predominantly to adherens junctions in placodal cell (Supplementary Fig. 5a–c). However, from early stage 11 onwards it was strongly localised in the apical-medial position, where it now colocalised with microtubule minus ends (Fig. 7d, e)[8]. This developmental relocalisation of Patronin accumulation depended on an intact microtubule cytoskeleton, as its apical-medial localisation in the placodal cells reverted to a junctional localisation upon depletion of microtubules via Spastin overexpression within the salivary gland placode only (using fkhGal4; Fig. 7f–h). Furthermore, the recruitment of Patronin also depended on the severing action of Katanin: when Katanin 80-YFP was degraded in the salivary gland placode using fkh-Gal4 UAS-degradFP, a Patronin-RFP transgene was mainly localised at apical junctions (Supplementary Fig. 5h–h″), whereas in the control it accumulated in the apical-medial position (Supplementary Fig. 5g–g″ and i). We further tested the severing-induced binding of Patronin to microtubule minus-ends by employing laser-induced severing of apical microtubules in early stage 11 (Fig. 7j) or later stage 12 (Fig. 7k) placodes compared to interphase (Fig. 7i) or mitotic (Supplementary Fig. 5l) epidermal cells as controls. This artificial severing led to a very rapid recruitment of Patronin-RFP to the new microtubules minus-ends generated by the ablation (Fig. 7i–m, Supplementary Fig. 5j–l and Supplementary Movies 3–6).

Thus, we identified Patronin as one factor that binds the minus ends of the longitudinal microtubule array in placodal cells, once microtubules were released by Katanin via severing from the nucleating centrosome.

**Loss of Patronin function prevents the proper organisation of the non-centrosomal microtubule array in the placode and apical-medial actomyosin recruitment.** Patronin is required during oogenesis[18] and therefore we could not generate maternal/zygotic mutants that would lack all Patronin protein in the embryo. Zygotic loss of Patronin alone only led to weakly penetrant phenotypes in salivary gland tube budding, most likely

due to a rescue by perdurance of maternal RNA and protein. Also, Patronin in the apical-medial position of placodal cells bound to microtubule minus-ends appeared to be very stable as several approaches, such as targeting Patronin by RNAi (Fig. 8) or tissue-specific degradation using a degradFP approach (Supplementary Fig. 6), only led to a reduction in Patronin protein levels but not a complete loss. To test Patronin requirement and reduce its levels, we used expression of an RNAi construct against patronin mRNA (UAS-Patronin-RNAi) under the control of a ubiquitous embryonic driver, daughterless-Gal4, DaGal4, (Fig. 8), thereby also bypassing an earlier requirement for Patronin during gastrulation[6]. In placodes of UAS-Patronin-RNAi x DaGal4 embryos (Fig. 8a–d″) medial Patronin fluorescence intensity was reduced by 24.4% (Supplementary Fig. 6c), whereas junctional Patronin intensity in placodes was reduced only by 14.9% (Supplementary Fig. 6d) compared to control embryos. Although fluorescence intensity of acetylated α-tubulin staining was not overall reduced within the whole apical domain of placodal cells (Supplementary Fig. 6a″, b″ and e), there was a clear change in organisation of the microtubule array (Fig. 8b–b′ versus d–d′). At stage 11 in the control, apical foci of longitudinal microtubule bundles were visible within the apical domain of cells close to the invagination point (Fig. 8b, b′), colocalising with medial Patronin foci (Fig. 8b″). Instead, when medial Patronin was lost due to Patronin-RNAi, microtubules were observed lying within the apical domain, often terminating in close proximity to junctions (Fig. 8d, d′), where Patronin was still localised (Fig. 8d″).

Concomitant with the loss of microtubule rearrangement, placodal cells of UAS-Patronin-RNAi x DaGal4 embryos showed a reduction in apical-medial F-actin accumulation compared to control embryos (Fig. 8e–g). This coincided with a disrupted morphology of these placodes (Fig. 8a″ versus c″). Furthermore, apical constriction appeared less efficient when apical-medial Patronin levels were reduced (Fig. 8h–j′). Hence, Patronin protecting the longitudinal arrangement of microtubules was important for the role of microtubules in supporting apical-medial actomyosin that we demonstrated previously[8]. We previously demonstrated that the spectraplakin and cytolinker Shot localises at the interface between microtubules and actomyosin within the apical-medial region of placodal cells, and affecting Shot function also leads to reduced apical

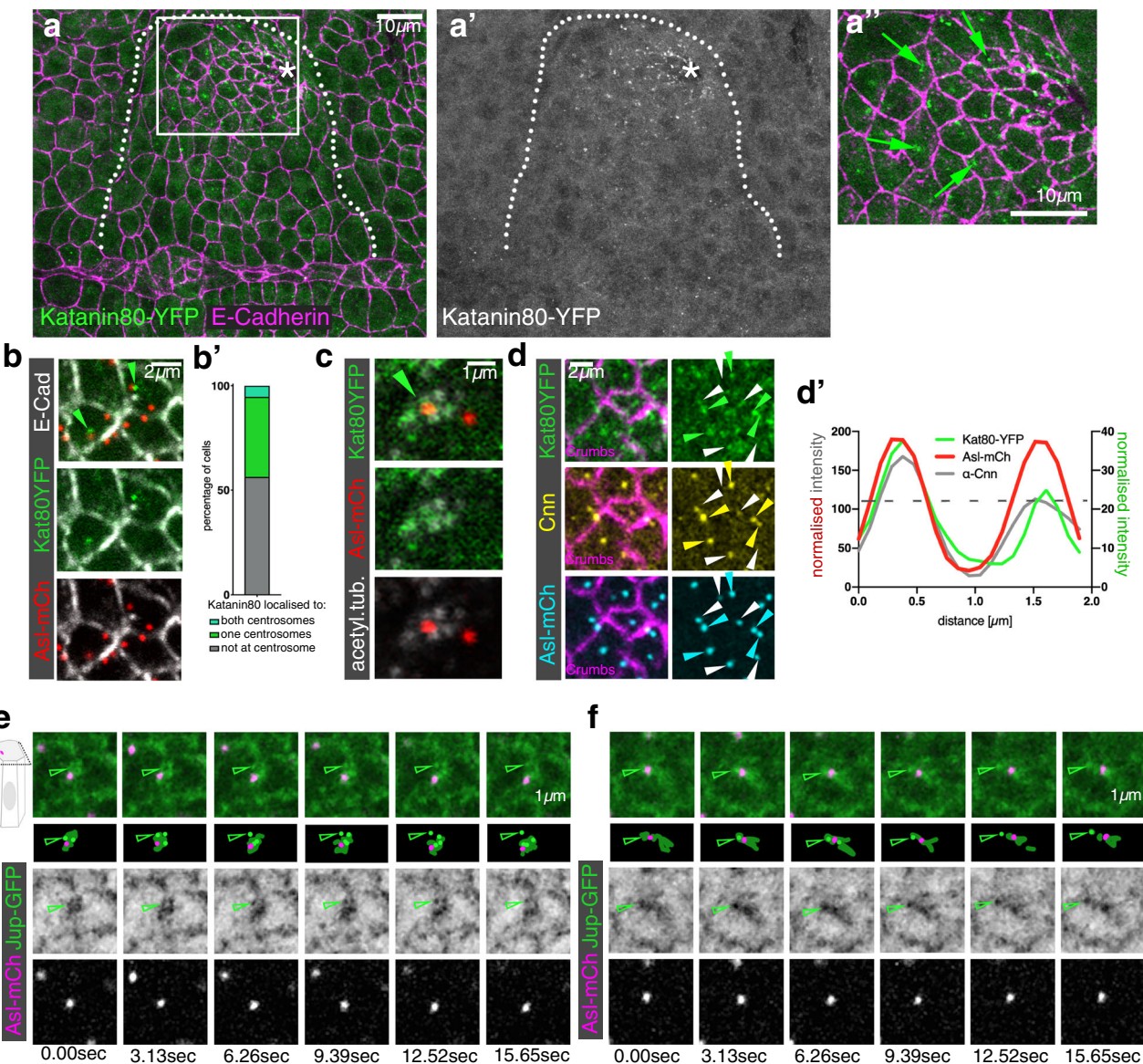

**Fig. 5 The microtubule-severing protein Katanin specifically accumulates at the apical-medial side of placodal cells. a–a″** Katanin80, labelled using a YFP-protein trap line (green in **a**, **a″** and single channel in **a**′), accumulates specifically in the secretory placodal cells and not the surrounding epidermis. **a″** Katanin80-YFP foci are found in an apical-medial position in the constricting population of cells (green arrows). Cell outlines are marked by E-Cadherin (magenta in **a**, **a″**). The salivary gland placode is indicated by a white dotted line and the invagination point by an asterisk. **b, c** Katanin80-YFP accumulates near centrosomes and microtubules. **b** Kat80YFP in green localises close to centrosomes marked by Asl-mCherry in red, cell outlines in white are labelled by E-Cadherin. **b**′ Quantification of Katanin80-YFP accumulation near centrosomes. **c** Katanin80-YFP (green) can be found near microtubules labelled by staining for acetylated α-tubulin (white) in the proximity of centrosomes marked by Asl-mCherry (red). **d–d**′ Katanin80-YFP (green) accumulates near the microtubule-nucleating centrosomes marked by Cnn (yellow), with all centrosomes marked by Asl-mCherry (cyan) and cell outlines marked by Crumbs (magenta). Solid arrowheads indicate centrosomes exhibiting stronger Cnn and Katanin accumulation, while hollow arrowheads point towards centrosomes with less or no Cnn and Katanin staining. **d**′ Line scan profile through both centrosomes of a single cell to illustrate the co-enrichment of Katanin80-YFP on the Cnn-enriched centrosome. Centrosome positions are marked by Asl-mCherry. **e, f** Live imaging of microtubules (labeled with Jupiter-GFP, a microtubule-binding protein, in green) and centrosomes marked by Asl-mCherry (magenta) reveals microtubule release from centrosomes, two examples are shown. Arrowheads point to Jupiter foci that lose association with the centrosome over the course of the movie.

constriction of the cells[8]. Patronin and Shot colocalised in apical-medial foci in placodal cells (Supplementary Fig. 6h–h″). When Patronin was reduced in these placodal cells via RNAi the apical-medial accumulation of not only Patronin but also of the cytolinker Shot was reduced (Supplementary Fig. 6i–k). Thus, Shot binding to Patronin could be an important link to recruit apical-medial F-actin through Shot's actin-binding domain.

As was observed when microtubules were lost within the placode[8], reduction of Patronin by RNAi also led to aberrant gland and lumen phenotypes at later stages of the tube invagination (Supplementary Fig. 7), demonstrating that the reorganisation of the microtubule array is required for efficient morphogenesis. Such aberrations in comparison to controls were also observed when Katanin80 was degraded (Supplementary Fig. 7), further suggesting an involvement in the same microtubule-rearrangement pathway. It also confirms that interference with wild-type patterns of apical constriction does not abolish tube budding ability, but rather leads to aberrant cell

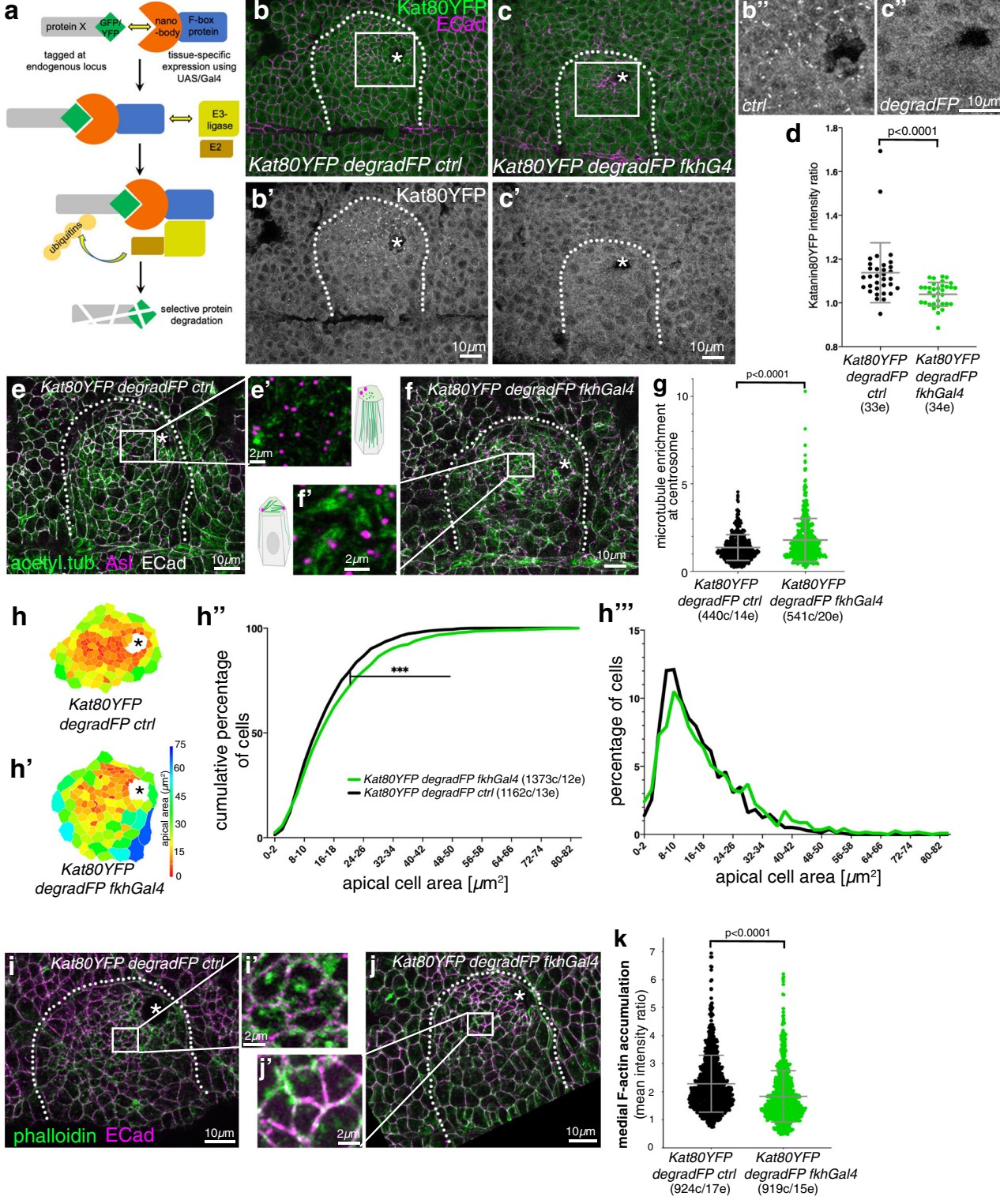

internalisation with resulting organ and cell shape defects. Hence, also γ-tubulin overexpression, which interfered with the usual patterning of apical constriction and internalisation (Fig. 4f–g), resulted in similar gland and lumen phenotypes at later stages (Supplementary Fig. 7).

In summary, Patronin in the salivary gland placode, once localised to microtubule minus-ends within the apical-medial domain, appears to serve to support the reorganisation of microtubules into a longitudinal array. We suggest this array in turn links to the apical-medial actomyosin via the cytolinker Shot,

and this arrangement is required for successful apical constriction of cells and formation of the wild-type tubular organ.

## Discussion

The organisation of the microtubule cytoskeleton is key to many cellular functions, both in individual cells as well as cells in the context of a tissue. In many actively cycling cells the interphase microtubule cytoskeleton is organised from centrosomes as the major MTOCs. In contrast, in many differentiated cells

**Fig. 6 Loss of Katanin disrupts the non-centrosomal microtubule array and apical-medial actomyosin-mediated apical constriction. a** Schematic of the 'degradFP' tissue-specific-degradation system (as in ref. [39]): tissue-specific expression of an F-box/anti-GFP-nanobody fusion protein, degradFP, (using UAS/Gal4)[71] leads to tissue-specific degradation of any endogenously GFP/YFP-tagged protein, in this case Katanin80-YFP. **b–d** Expressing degradFP using *fkh-Gal4* in the salivary gland placode (*Kat80YFP degradFP fkhG4*; **c–c"**) leads to significant loss of Katanin80-YFP compared to control (*Kat80YFP degradFP ctrl*; **b–b"**). Cell outlines are marked by E-Cadherin (E-Cad). **d** Quantification of Katanin80-YFP depletion (*Kat80YFP degradFP ctrl* $n = 33$ embryos; *Kat80YFP degradFP fkhG4* $n = 34$ embryos; shown are mean ± SD, statistical significance was determined by two-sided unpaired Mann–Whitney test as $p < 0.0001$). **b"** and **c"** are higher magnifications of the white boxes marked in **b** and **c**, respectively. **e–g** In placodes where Katanin80-YFP is degraded (**f, f'**), microtubules (green, labelled for acetylated α-tubulin) remain localised within the apical domain and in contact with centrosomes (magenta, labeled for Asl) compared to control (**e, e'**) where a non-centrosomal longitudinal array is formed. **e'** and **f'** are magnifications of the areas indicated in **e** and **f** by a white box. **g** Quantification of the effect shown in **e, f**; (*Kat80YFP degradFP ctrl*: 440 cells from 14 embryos; *Kat80YFP degradFP fkhG4*: 541 cells from 20 embryos; shown are mean ± SD, statistical significance was determined by two-sided unpaired Mann–Whitney test as $p < 0.0001$). **h-h"** Katanin80-YFP degradation (**h'**) leads to a loss of apical constriction compared to control (**h**), apical area of cells of example placodes are shown in a heat map. **h"** Quantification of apical area distribution of placodal cells in control (ctrl) and Katanin depleted (*degradFP fkhGal4*) placodes at stage 11 showing the cumulative percentage of cells relative to apical area size. **h"'** Percentage of cells in different size-bins [Kolmogorov-Smirnov two-sample test, $p \ll 0.001$ (***)]. 12 placodes were segmented and analysed for control and 13 for Katanin80-YFP depletion, the total number of cells traced was N(*Kat80YFP degradFP ctrl*)=1373, N(*Kat80YFP degradFP fkhG4*) = 1162. **i–k** In placodes where Katanin80-YFP is degraded (**j, j'**), apical-medial F-actin (green, labeled using phalloidin) is reduced compared to control (**i, i'**) where apical-medial actin is highly prevalent. **i'** and **j'** are magnifications of the areas indicated in **i** and **j** by a white box. **k** Quantification of loss of apical-medial F-actin (*Kat80YFP degradFP ctrl*: 924 cells from 17 embryos; *Kat80YFP degradFP fkhG4*: 919 cells from 15 embryos); shown are mean ± SD, statistical significance was determined by two-sided unpaired Mann–Whitney test as $p < 0.0001$. The salivary gland placode is indicated by a white dotted line and the invagination point, where present, by an asterisk.

microtubules are nucleated or anchored from sites independent of centrosomes to support specific cellular functions. This is true for most epithelia and neurons in animals but also yeast and plant cells.

Despite non-centrosomal microtubule arrays being common features in such differentiated cells, the mechanisms of their generation and organisation are still unclear[36]. On the one hand, non-centrosomal microtubules can be de novo nucleated at non-centrosomal MTOCs, and for this process γ-tubulin as part of the γ-TURC is often essential[12,43]. However, the mechanisms leading to recruitment of γ-tubulin at non-centrosomal MTOCs is still poorly understood. For example, a splice variant of Cnn, CnnT, resides at mitochondria in *Drosophila* spermatids to recruit γ-tubulin and convert the mitochondria into MTOCs[44]. However, de novo nucleation at non-centrosomal MTOCs is only one way of generating a non-centrosomal array. Rather than relocalising the nucleation capacity away from centrosomes, a release and capture of microtubules generated at nucleating centrosomes is an alternative, though not mutually exclusive, mechanism to form non-centrosomal microtubule arrays that has been first observed in neurons[45]. Furthermore, once even a small cluster of non-centrosomal microtubules has formed, this organisation can be selectively amplified by targeted severing of such microtubules combined with capture of newly generated minus ends and continued polymerisation from free plus ends.

Tube budding morphogenesis of the salivary glands in the *Drosophila* embryo, and in particular the apical constriction of cells leading to tissue bending, depends on an intact microtubule cytoskeleton that is organised as a non-centrosomal longitudinal array[8]. Here we elucidate the mechanism by which this non-centrosomal microtubule array is formed. Interestingly, the initial formation of the array still involves centrosomal nucleation, though in the salivary gland placodal cells this is restricted to the Centrobin-enriched and hence possibly daughter centrosome that retains nucleation capacity after the last embryonic division. Both non-centrosomal array formation as well as the downstream morphogenetic process of apical constriction also require the action of the severing enzyme Katanin[38], as well as the minus-end stabiliser Patronin[34,35]. Thus, in this tissue a mechanism appears to operate whereby at least initially centrosomally nucleated microtubules are severed by Katanin at the centrosome and their free minus ends then recruit Patronin. Minus ends bound to Patronin remain anchored within the apical-medial region[8]

where they promote actomyosin recruitment or stabilisation through a binding partner also localised to this hub, the spectraplakin Shot[8].

It is curious to speculate what the close apposition and interaction of microtubules, and in particular their minus ends, and apical-medial actomyosin in apically constricting cells entails. Our past and recent data strongly suggest a regulatory interplay, with the presence and amount of non-centrosomal microtubules directly affecting presence and amount of apical-medial actomyosin and thus the rate of apical constriction in placodal cells. The loss of microtubules leads to loss of apical-medial myosin and hence reduction in apical constriction[8], and increase in placodal non-centrosomal microtubules leads to an increase in apical-medial actomyosin and increased apical constriction, as described above. Shot, as one of the largest proteins encoded in most animal genomes, provides ample potential binding sites for regulatory proteins that could impinge on actomyosin activity. Many questions remain, such as whether this interaction is regulatory in that microtubule minus ends near apical-medial actomyosin could serve to guide directional transport of vesicles to the apical domain, and recent evidence suggests that this might at least partially be the case[46]. There could be a mechanical requirement for a close apposition of longitudinal microtubules and apical-medial actomyosin in allowing the formation of a wedged shape of an apically constricting cell, akin to ideas about cellular 'tensegrity' originally proposed by Ingber et al.[47]. How the latter could be tested experimentally is not clear, though in silico modelling might pave a way for a better understanding of mechanical implications of this apposition.

Interestingly, a different requirement for microtubules in assisting actomyosin-based apical constriction was recently reported during mesoderm invagination in the *Drosophila* embryo. Here, disrupting microtubules through colchicine or taxol injections to either depolymerise or stabilise the network acutely led to longer persistence and size of the interconnected apical-medial actomyosin network that stretches across cells[6]. Microtubules appear necessary to induce the actin turnover near junctions where the apical-medial foci connect via cell–cell adhesions between neighbouring cells.

Also in the early embryo during gastrulation, Patronin appears to behave more dynamically and affected microtubules in a different way, suggesting that roles for Patronin might be developmental-stage specific. Expression of shRNA targeting

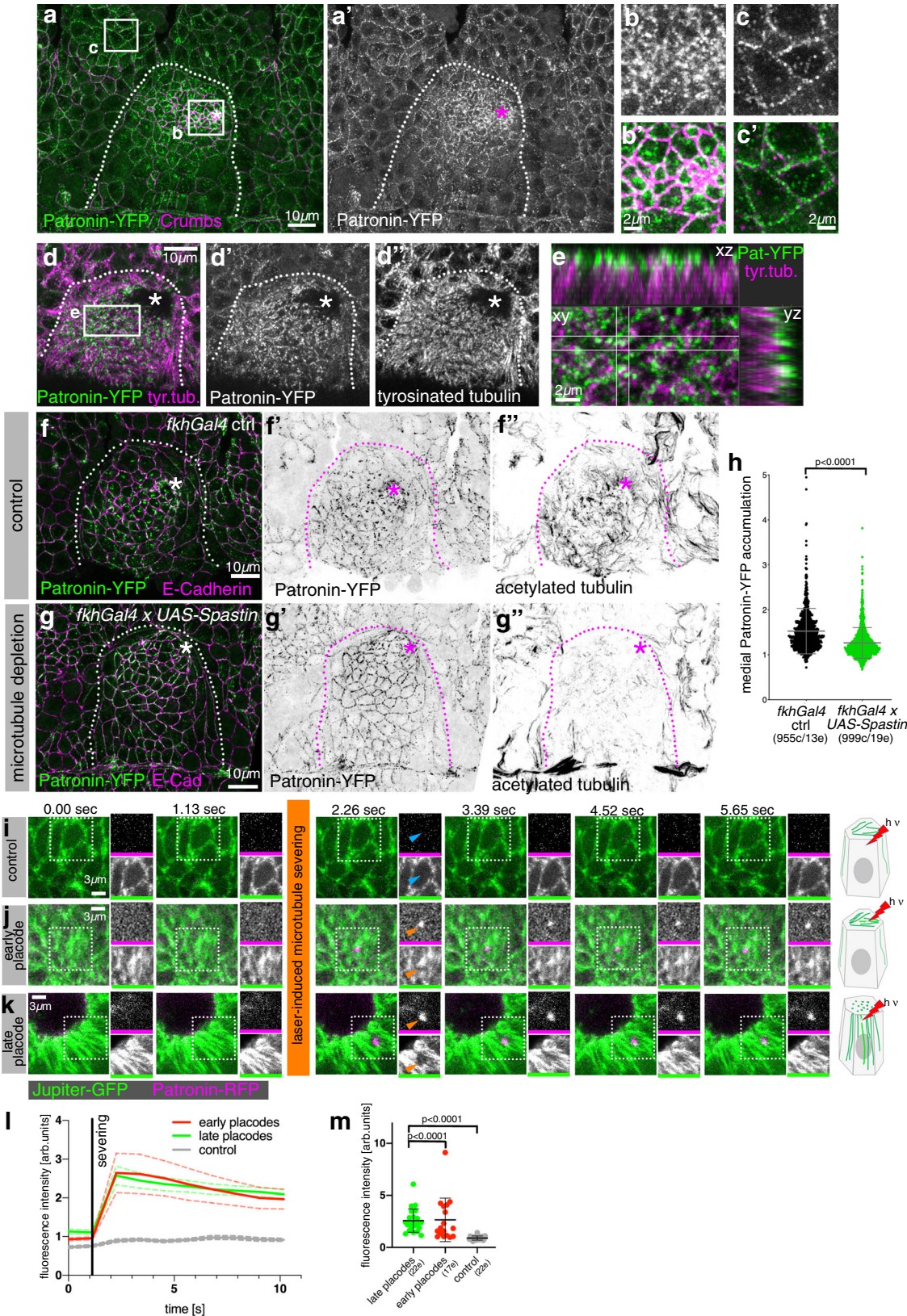

Patronin at this early stage led to a strong overall loss of Patronin, and a concomitant loss of acetylated microtubules[6]. Several hours further into embryogenesis, during the formation of the salivary gland tubes analysed here, Patronin is more stable and resistant to depletion (Fig. 8 and Supplementary Fig. 6) and its loss or reduction does not lead to concomitant loss but rather

disorganisation of microtubules. Within the placode, the reduction in apical-medial Patronin without overall loss of microtubules could be due to junctional Patronin now binding microtubules that were severed at centrosomes and require anchoring. Alternatively, as has been described in cells in culture, loss or reduction of Patronin/CAMSAPs can directly affect the

**Fig. 7 Patronin localises specifically to non-centrosomal microtubule minus ends in the apical-medial region of placodal cells. a–c′** Patronin-YFP (green and **a′**, **b**, **c**), accumulates at adherens junctions (magenta) throughout the embryonic epidermis (**a** and **c**, **c′**), but in the apically constricting cells of the placode accumulates in an apical-medial position (**a** and **b**, **b′**). **b** and **c′** are magnification of the white boxes in **a**. **d**, **e** Apical-medial Patronin-YFP localises to the minus-ends of longitudinal microtubules labeled for tyrosinated α-tubulin. **e** is a magnification of the white box in **d** and also shows corresponding xz and yz-cross-sections. **f–h** Patronin depends on microtubules for its apical-medial localisation. In contrast to control (**f–f″**) where Patronin-YFP localises to apical-medial sites, when microtubules are lost upon expression of *UAS-Spastin* under *fkh-Gal4* control (**g–g″**), Patronin-YFP continues to be localised to junctional sites and does not relocalise to apical-medial sites. **h** Quantification of reduction of apical-medial Patronin-YFP upon placodal microtubule loss (*fkhGal4 ctrl*: 955 cells from 13 embryos; *fkhGal4 x UAS-Spastin*: 999 cells from 19 embryos/ shown are mean ± SD, statistical significance was determined by two-sided unpaired Mann–Whitney test as p < 0.0001). **i-m** Stills of time lapse movies of laser-induced microtubule ablation. Compared to a control cut just below the apical microtubules (**i**), laser-induced ablation of microtubules within the apical microtubule array in early stage 11 placodes (**j**) as well as in the apical region of the longitudinal microtubule array in later stage 12 placodes (**k**) leads to a very rapid recruitment of Patronin-RFP to the newly generated minus-ends of microtubules. Orange arrowheads in **j**, **k** point to the severed site and Patronin-RFP recruitment, blue arrows in **i** point to the control cut position. A *Jupiter-GFP Patronin-RFP* genotype was analysed in all instances. The dotted boxes indicate the area shown in individual black and white panels. See also Supplementary Fig. 5 and Supplementary Movies 3–6. **l** Time-resolved quantification of the fluorescence intensity of Patronin-RFP at the site of laser-ablation in early and late placodes compared to control. Mean ± SEM are shown, n = 17 (early placodes), n = 22 (late placodes), n = 22 (control). **m** Quantification of Patronin-RFP intensity in the first image acquired post-ablation in early and late placodes and control. Shown are mean ± SD; n = 17 (early placodes), n = 22 (late placodes), n = 22 (control); statistical significance was determined by two-sided unpaired Mann–Whitney test as p < 0.0001 where indicated. The salivary gland placode is indicated by dotted lines and the invagination point by asterisks.

---

ability of Katanin to sever microtubules[48], and thus upon Patronin depletion more microtubules may remain attached to the nucleating centrosome. Supporting this, Patronin colocalises with a pool of Katanin in the secretory cells of the placode (Supplementary Fig. 6f, g‴). Such a disorganisation of microtubules upon perturbance of Patronin is also more reflective of its behaviour in the adult epithelium of the follicle cells in the fly ovary[49] and the small intestine of postnatal mice[9]. This most likely reflects changes from actively dividing to post-mitotic epithelial behaviour in both tissues, and is also supported by observed changes to CAMSAP3 mobility during maturation of Caco2 cell cysts in culture[50].

Thus, it appears that depending on the tissue context, including the mitotic activity that epithelial cells display at a given time, the interplay of centrosome-nucleated microtubules, microtubule-release from centrosomes, as well as the function of CAMSAP-family proteins such as Patronin are highly coordinated and adjusted to serve the assembly and maintenance of particular microtubule arrays. In the case of the formation of the tubular organ of the salivary glands, a model process for tube formation, we have elucidated a key mechanism that harvests the changes at centrosomes due to the cells becoming post-mitotic (Fig. 8k) and pairs it with the activity of Katanin and Patronin to promote formation and maintenance of the longitudinal non-centrosomal microtubule array that itself supports apical constriction of cells. It will be interesting to determine in the future how conserved this interplay is in other tissues reaching post-mitotic state but still undergoing morphogenetic processes such as apical constriction and tissue bending.

## Methods

**Drosophila stocks and genetics.** The following fly stocks were used in this study: from Bloomington Stock Centre: *Daughterless-Gal4* (*Da-Gal4*; #27608); *UAS-Patronin-RNAi* (y1 sc* v1; P{TRiP.HMS01547}attP2) #36659); *ncd-γ-tubulin-EGFP* (*w1118*; P{ncd-γTub37C.GFP}F13F3)(#56831). Furthermore *Katanin80-YFP* (*w1118 PBac{602.P.SVS-1}kat80CPTI000764*) [Kyoto Stock Centre; CPTI 000764]; *fkh-Gal4* on chromosome III[51,52]; *Asterless-GFP on X, Ubi-EB1-mCherry on X* and *Asterless-mCherry* (*w; eAsl-mch/Cyo; MKRS/TM6b*) [gifts form Jordan Raff]; *YFP-Cnb* (*w1118; pUbi-YFP-Cnb*)[29]; *RFP-Cnn*[30]; *Ubi-EB1-GFP*[53]; *Patronin-RFP* (*Patronin-TagRFPattp40[22H02-C]*)[54]; *Patronin-YFP* (*w1118; Patronin-YFP/Cyo*)[18]; *GFP-Polo*[55]; *Sas-4-GFP*[56]; *Scribble-GFP* (*w; P{PTT-GA}scribCA07683*)[57]; *Spd-2-GFP*[58]; *sqh-TagRFPt[9B]*[33]; *UAS-deGradFP* (*w; If/Cyo; UAS>NSlmb-vhhGFP4/TM6b*)[39]; *UAS-Spastin on X*[59]; *Jupiter-GFP* (P{PTT-GA}JupiterG00147)[60]; *fkh-Gal4 UAS-srcGFP*[61].

The following combinations of transgenes were generated in this study:
*Asl-mCherry; Jupiter-GFP*
*EB1-mCherry;; ncd::γ-tubulin-EGFP*
*sqh-TagRFPt[9B];; ncd::γ-tubulin-EGFP*

*Katanin80-YFP; Asl-mCherry*
*Katanin80-YFP;; fkh-Gal4*
*Katanin80-YFP;; UAS-degradFP*
*Katanin80-YFP; Patronin-RFP; fkh-Gal4*
*Katanin80-YFP; Patronin-RFP; UAS-degradFP*
*Patronin-RFP; ncd::γ-tubulin-EGFP*
*Patronin-RFP; Jupiter-GFP*
*Patronin-YFP; Da-Gal4*
*Patronin-YFP; fkh-Gal4*
*Patronin-YFP; UAS-degradFP*
*UAS-spastin; Patronin-YFP*
*Patronin-YFP; Ubi-TagRFP*

Genotypes analysed are indicated in the figure panels and legends, and are also described under Fly husbandry below.

**Fly husbandry.** To analyse cell boundaries compared to centrosome components, *Asl-mCherry* flies were crossed to *Scribble-GFP*. Analysis of YFP-Cnb and Cnn enrichment on centrosomes was done by crossing *YFP-Cnb* males with *Asl-mch* virgins. To analyse microtubule nucleation from Cnn-positive centrosomes, *RFP-Cnn* flies were crossed to *Ubi::EB1-GFP* flies.

Imaging of EB1 comets between control embryos and γ-tubulin-overexpressing embryos was done the following way: control embryos were obtained by crossing *EB1-mCherry* on X virgins with *Asl-GFP* on X males. EB1 comets were analysed in parallel in the strain *EB1-mCherry;; ncd::γ-tubulin-EGFP*.

To degrade Katanin80-YFP specifically in the salivary gland placode, virgins of the genotype *Katanin80-YFP fkhGal4* were crossed to males of the genotype *Katanin80-YFP UAS-degradFP III*, so that all offspring was homozygous for *Katanin80-YFP*. *Katanin80-YFP degradFP III* was analysed as a control.

To analyse Patronin localisation when Katanin80-YFP was degraded specifically in the salivary gland placode, virgins of the genotype *Katanin80-YFP; Patronin-RFP; fkhGal4* were crossed to males of the genotype *Katanin80-YFP; Patronin-RFP; UAS-degradFP*, so that all offspring was homozygous for *Katanin80-YFP*. *Katanin80-YFP; Patronin-RFP; degradFP* was analysed as a control.

To analyse Patronin-YFP localisation when microtubules were depleted, virgins of *Patronin-YFP; fkhGal4* were crossed to males of *UAS-Spastin (X); Patronin-YFP*. Using anti-tubulin immunofluorescence, successful MT-depletion was confirmed and only these placodes analysed.

Degradation of Patronin-YFP via the degradFP approach was done by crossing *Patronin-YFP; DaGal4* virgins with *Patronin-YFP; UAS-degradFP* males.

To analyse loss of Patronin in embryos, *UAS-Patronin-RNAi* was expressed throughout the early embryo using *daughterlessGal-4* (*DaGal4*) at 29 °C. *UAS-Patronin-RNAi* virgins were crossed to *Patronin-YFP; DaGal4* males. Control embryos were obtained by crossing *Patronin-YFP; Da-Gal4* males with *yw* virgins. Only embryos where Patronin-YFP staining reduction was visible were analysed.

**Embryo immunofluorescence labelling, confocal, and live analysis.** Embryos were collected on apple juice-juice plates and processed for immunofluorescence using standard procedures. Briefly, embryos were dechorionated in 50% bleach, fixed in 4% MeOH-free formaldehyde, and devitellinised in a 50% mix of 90% EtOH and Heptane. They were then stained with phalloidin or primary and secondary antibodies in PBT (PBS plus 0.5% bovine serum albumin and 0.3% Triton X-100). Anti-E-Cadherin (DCAD2, 1:10 dilution), anti-CrebA (CrebA Rbt-PC, 1:1000) and anti-Crumbs (Cq4, 1:10) antibodies were obtained from the

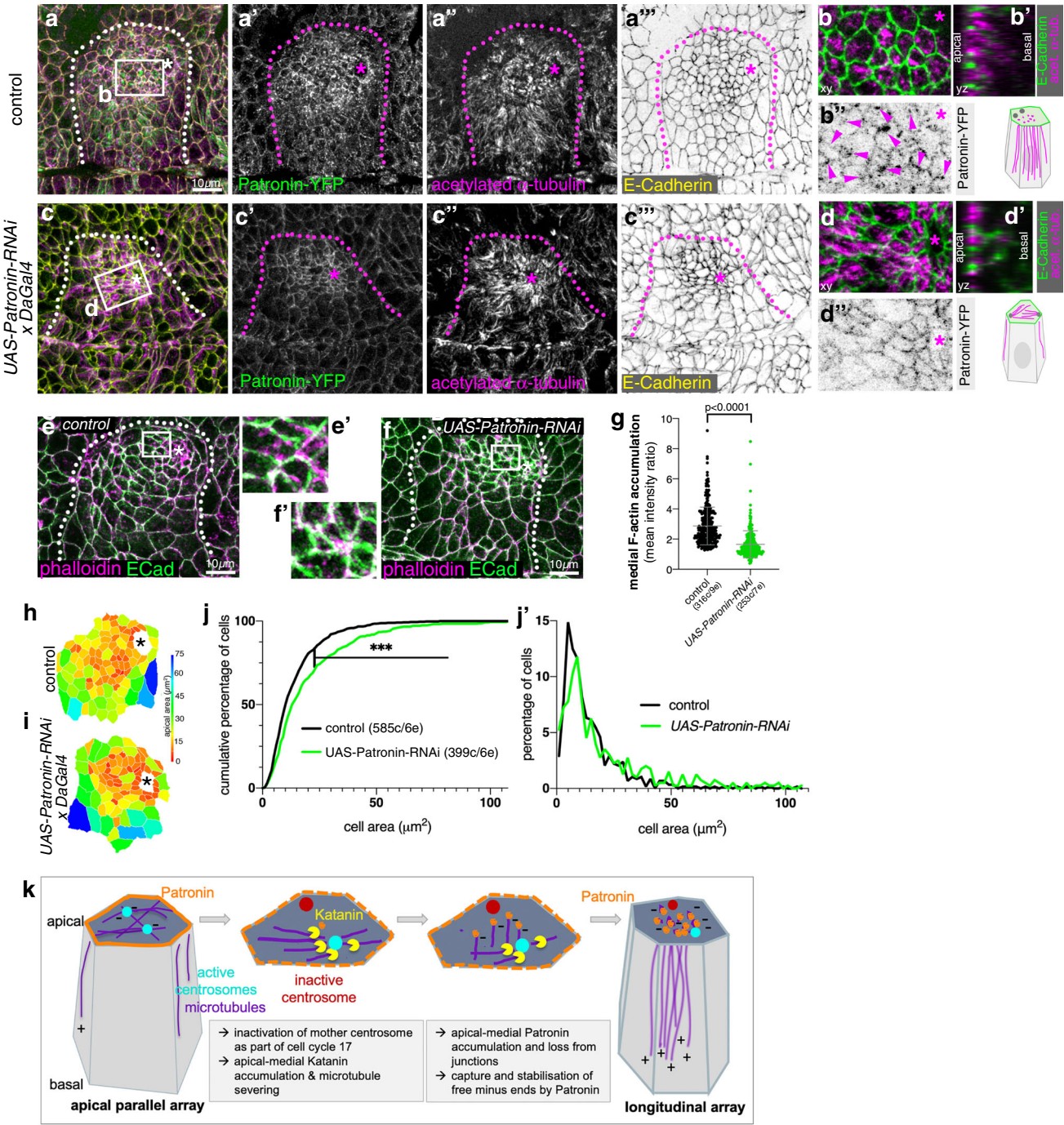

Developmental Studies Hybridoma Bank at the University of Iowa; anti-aPKC (sc-216; Santa Cruz) anti tyrosinated α-tubulin (YL1/2, 1:10)[62]; anti-acetylated α-tubulin (clone 6-11B-1, 1:500; Sigma); anti-α-tubulin (DM1A, 1:1000; Sigma); anti-γ-tubulin (clone GTU-88, 1:500; Sigma); anti-phospho-Histone H3 [Ser10] (Cell Signalling Technology; #9701, 1:500); anti-Asterless-NT; and anti-Cnn[63] (1:1000n for both) were a kind gift from Jordan Raff[64]; anti-Shot[65] (1:1000). The following secondary antibodies were used at 1:200: Alexa Fluor® 488 AffiniPure Donkey Anti-Rabbit IgG (H + L) (711-545-152); Cy™3 AffiniPure Donkey Anti-Rabbit IgG (H + L) (711-165-152); Alexa Fluor® 647 AffiniPure Donkey Anti-Rabbit IgG (H + L) (711-605-152); Cy™3 AffiniPure Donkey Anti-Mouse IgG (H + L) (715-165-151); Cy™5 AffiniPure Donkey Anti-Mouse IgG (H + L) (715-175-151); Cy™3 AffiniPure Goat Anti-Rat IgG (H + L) (112-165-167); Alexa Fluor® 647 AffiniPure Donkey Anti-Rat IgG (H + L) (712-605-153); Cy™5 AffiniPure Donkey Anti-Guinea Pig IgG (H + L) (706-175-148) were from Jackson ImmunoResearch Laboratories. Donkey anti-Rabbit IgG (H + L) Highly Cross-Adsorbed Secondary Antibody, Alexa Fluor Plus 405 (A48258); Goat anti-Mouse IgG (H + L) Highly Cross-Adsorbed Secondary Antibody, Alexa Fluor 350 (A-21049); Donkey anti-Mouse IgG (H + L) Highly Cross-Adsorbed Secondary Antibody, Alexa Fluor Plus

488 (A32766); Goat anti-Rat IgG (H + L) Cross-Adsorbed Secondary Antibody, Alexa Fluor 488 (A-11006) were from Invitrogen, and rhodamine-phalloidin (1:500) was from Thermofisher (R415). Samples were embedded in Vectashield (Vectorlabs H-1000).

Images of fixed samples were acquired on an Olympus FluoView 1200 (with the FV10-ASW v04.02 software) or a Leica SP8 inverted microscope (LAS X software) equipped with 405 nm laser line for four-colour imaging as z-stacks to cover the whole apical surface of cells in the placode. z-stack projections were assembled in ImageJ or Imaris (Bitplane), 3D rendering was performed in Imaris.

For live time-lapse imaging, the embryos were dechorionated in 50% bleach, rinsed in water and attached to a coverslip with the ventral side up using heptane glue and covered with Halocarbon Oil 27. Time-lapse sequences of Ubi::EB1-GFP RFP-Cnn were acquired on a Leica SP8 inverted microscope (63x/1.4NA Oil objective; LAS X v3.5.2.18963 software) as z-stacks, while Patronin-YFP; Ubi-TagRFP was imaged on a Zeiss 780 inverted microscope (with Zen 2.1 SP3 FP2 v14.0.16.201software) with a ×40/1.3NA Oil objective as a single confocal slice, using linear unmixing to remove the background fluorescence of the embryonic vitelline membrane. Similarly Asl-GFP EB1-mCherry, EB1-mCherry γ-tubulin and

**Fig. 8 Loss of Patronin affects microtubule organisation and apical constriction in the placode. a–d**″ Patronin-YFP (**a′**, **b**″, **c′**, **d**″ and green in **a**, **c**) and acetylated α-tubulin (**a**″, **c**″ and magenta in **a**, **b**, **b′**, **c**, **d**, **d′**) labelling in control embryos (**a**, **b**″) and *UAS-Patronin-RNAi x DaGal4* embryos (**c**, **d**″). E-Cadherin to label cell outlines is in yellow in **a**, **b**, green in **b**, **b′**, **d**, **d′**. **b′** and **d′** are z-sections of cells at the positions of the dotted lines indicated in **b** and **d**. See quantification in Supplementary Fig. 6. **e–g** Depletion of Patronin using RNAi in *UAS-RNAi-Patronin x DaGal4* embryos (**f**, **f′**) leads to loss of apical-medial F-actin labelled using phalloidin (magenta) in contrast to control (**e**, **e′**). **e′** and **f′** are magnifications of the white boxes shown in **e** and **f**. Membranes are labelled by E-Cadherin (green). **g** Quantification of changes of apical-medial F-actin accumulation upon Patronin depletion; *control:* 316 cells from 9 embryos; *UAS-RNAi-Patronin x DaGal4:* 253 cells from 7 embryos; shown are mean ± SD, statistical significance was determined by two-sided unpaired Mann–Whitney test as $p < 0.0001$. **h–j′** Depletion of Patronin in *UAS-Patronin-RNAi x DaGal4* embryos (**i**) leads to a loss of apical constriction compared to control (**h**), apical area of cells of example placodes are shown in a heat map. **j**, **j′** Quantification of apical area distribution of placodal cells in control and Patronin-depleted (*UAS-Patronin-RNAi x DaGal4*) placodes at stage 11 showing the cumulative percentage of cells relative to apical area size (**j**) as well as the percentage of cells in different size-bins (**j′**) [Kolmogorov–Smirnov two-sample test, $p \ll 0.001$ (***)]. Six placodes were segmented and analysed for each condition, the total number of cells traced was $N(control) = 585$, $N(UAS-Patronin-RNAi x DaGal4)=399$. **k** Model of generation of the longitudinal non-centrosomal microtubule array in salivary gland placodal cell prior to apical constriction: inactivation of the Centrobin-depleted centrosome concomitant with entering cycle 17 and becoming post-mitotic leads to restriction of microtubule nucleation to a single centrosome. An increase in apical-medial Katanin levels in the secretory cells drive severing of microtubules at the active centrosome. Severed microtubule minus-ends are then captured by apical Patronin in the apical-medial region, thereby promoting the longitudinal microtubule arrangement. In overview panels the salivary gland placode is indicated by a dotted line and the invagination point by an asterisk.

*Asl-mCherry Jupiter-GFP* movies were acquired in the same way but on a Zeiss 880 inverted microscope (Zen 2.3 SP1 FP3 v14.0.20.201 software). Images were acquired every 1.14 s for EB1 comets and every 3.13 s for Jupiter-GFP. Finally, laser ablation experiments were performed on a multiphoton Zeiss 710 NLO confocal microscope (Zen 2010 software, version v6.0.0.309) equipped with a Ti-sapphire laser. Images were acquired every 1.13 s, and the laser cut was performed after the second time point with the Ti sapphire laser at 800 nm set at 80% laser power with 1 iteration.

Z-stack projections to generate movies were assembled in ImageJ or Imaris.

### Quantifications

*Proportion of Cnn-positive centrosome nucleating microtubules.* Nucleation was manually assessed on time-projections of confocal images from time-lapse movies, assessing whether EB1-GFP comets were seen emanating from Cnn-positive centrosomes or not.

*Centrosome asymmetry index.* Centrosome intensities were measured for both centrosomes in every single cell in an area of 32.48 μm × 25.35 μm close to the invagination pit in late stage 11-early stage 12 embryos on z-projections of the most apical planes (1–3 μm depending on the orientation of each embryo). The fluorescence was measured in a circle of 0.68 μm diameter, aiming to cover the maximum area of a centrosome as determined by the area measured for the extension of γ-tubulin, the most outer PCM component of the centrosome[66]. The ratio was obtained by dividing the fluorescence intensity of the brighter of the two centrosomes by the fluorescence intensity of the other centrosome in any given cell. Numbers analysed are as follow: *Sas4-GFP* (5 placodes, 170 cells); *Spd2-GFP* (4 placodes, 139 cells); α-Asl (9 placodes, 302 cells); α-γ-tubulin (12 placodes, 487 cells); *GFP-Polo* (7 placodes, 270 cells); α-Cnn (9 placodes, 309 cells); *YFP-Cnb* (5 placodes, 157 cells); *ncd::γ-tubulin-EGFP* (8 placodes, 284 cells).

*MT intensity measurements.* The raw intensity of microtubule labelling in γ-tubulin-EGFP overexpression or Patronin-YFP depletion experiments (α-tubulin, anti-tyrosinated α-tubulin or anti-acetylated α-tubulin) was measured in the entire placode on z-projections of the most apical planes (1–3 μm depending on the orientation of each embryo).

*Automated cell segmentation and apical area analysis.* For the analysis of apical cell area, images of fixed embryos of late stage 11/early stage 12 placodes, labelled with DE-Cadherin to highlight cell membranes and with dCrebA to mark salivary gland fate, were analysed. Cells were segmented in confocal image stacks, with cell analysis software (otracks, custom software written in IDL, from L3 Harris Geospatial, https://www.l3harrisgeospatial.com/Software-Technology/IDL; code availabe on request from Dr. Guy Blanchard [gb288@cam.ac.uk]) as used and published previously[8,67–69]. Briefly, the shape of the curved placode surface was identified in each z-stack as a contiguous 'blanket' spread over the cortical signal. Quasi-2D images for cell segmentation containing clear cell cortices were extracted as a maximum intensity projection of the 1 or 1.5 μm-thick layer of tissue below the blanket. These images were segmented using an adaptive watershed algorithm. Manual correction was used to perfect cell outlines for fixed embryos. Only cells of the salivary placode were used in subsequent analyses and were distinguished based on dCrebA staining.

*Number of wavy junctions.* The fraction of junctions displaying waviness was determined manually by counting how many junctions within the secretory region of the analysed placodes ($n = 7$ placodes for control, $n = 10$ placodes for *ncd::γ-*

*tubulin-EGFP* embryos) out of the total number of junctions in that area displayed waviness, i.e. an undulating deviation from the direct line between two vertices.

*Straightness measurements of junctions.* This straightness of a junction was determined by dividing the ideal junction length between vertices ($L_{[direct\ route]}$) by the actual junction path length ($L_{[junction]}$, see Fig. 3K), based on the E-cadherin staining as in ref. [31]. $L_{[direct\ route]}$ and $L_{[junction]}$ were determined manually using FIJI. Only junctions considered as wavy junctions in the above quantification were quantified ($n = 90$) in *ncd::γ-tubulin-EGFP* embryos that overexpress γ-tubulin-EGFP. The same number of junctions was analysed in control placodes. As wavy junctions were also at times observed in control embryos and in order to achieve unbiased analysis, junctions to be quantified in control embryos were selected using an online random number generator.

*Katanin80-YFP accumulation at centrosomes.* The Katanin80-YFP protein trap strain displays a strong homogeneous background fluorescence. Due to this, images were processed as follows to be able to determine actual Katanin80-YFP fluorescence at centrosomes rather than measure the background fluorescence. The Katanin80-YFP channel was selected and subjected to background subtraction using the subtract background function in Fiji with a rolling ball radius of 50 pixels. The image was then smoothed and thresholded using the automatic Triangle method in Fiji to obtain a mask of the Katanin80-YFP channel. The pixel values of the mask were divided by 255 to generate a binary mask of Katanin80-YFP where Katanin signal corresponds to pixel value of 1. The binary mask was then used to multiply the pixel values of the original image by 0 or 1, to obtain the actual Katanin80-YFP staining corresponding to pixels with a value of 1 in the binary mask, and to therefore obtain the processed Katanin80-YFP image. Katanin80-YFP intensity was then measured in the processed image in an area of 33 × 21 μm close to the invagination pit, by drawing a circle ROI of 0.68 μm diameter surrounding each centrosome marked by Asl-mCherry. Results shown correspond to 232 cells in 7 embryos.

*Quantification of Katanin80-YFP.* Katanin80-YFP staining was measured in the placode and the surrounding epidermis (33 control embryos and 34 Katanin-depleted embryos were analysed in z-projections of the most apical planes, 1–3 μm depending on the orientation of each embryo). As Katanin80-YFP embryos displayed a hazy background fluorescence, we determined the background fluorescence from five ROIs deeper within the embryo where there is no bona fide Katanin fluorescence and subtracted this from the values obtained. We then calculated the ratio of Katanin80-YFP enrichment in the placode by dividing the intensity value in the placode by the intensity value determined in the surrounding epidermis. 33 control embryos and 34 Katanin80-depleted embryos were analysed.

*Acetylated α-tubulin accumulation near centrosomes.* The microtubule intensity was measured in an ROI of 0.68 μm diameter around the centrosomes (labelled with Asl-mCherry) on Z projections of the most apical planes (1–3 μm depending on the orientation of each embryo), and the intensity was normalised against the mean microtubule intensity in the entire placode.

*Medial accumulation of phalloidin, sqh-RFP, Patronin-YF, Patronin-RFP and Shot.* Images were taken of salivary gland placodes and surrounding tissue at late stage 11/early stage 12. Maximum intensity projections of the apical surface of placodal cells were generated using 3–5 optical section separated by 1 μm each in z. For each embryo analysed, fluorescence measurements were made for all secretory cells within the placode except cells close to the actomyosin cable. The medial and junctional values were measured after drawing the cells outlines (7-pixel wide line)

with a home-made plugin in Fiji, available on request. 10 cells were similarly analysed in the surrounding tissue.

For Patronin-YFP, Sqh-RFP and phalloidin quantifications, the graphs display the medial accumulation corresponding to the ratio between the medial intensity for each cell in the placode divided by the mean intensity of the 10 cells outside the placode. For the comparison of Patronin-YFP in the placode and the surrounding epidermis, the graph displays the ratio between the medial Patronin-YFP intensity versus the background intensity measured deeper within the embryo where there is no bona fide Patronin fluorescence. For Patronin-RFP and Shot, due to the noisy labeling in the surrounding epidermis, the graph displays the ratio between Patronin-RFP or Shot medial staining versus junctional staining for each cell in the placode.

*Laser ablation experiments.* Patronin-RFP fluorescence intensity was measured in a small ROI (3.09 μm × 3.09 μm) over 10 time points, and normalised for each time point against Patronin-RFP intensity throughout the entire field of view. A value higher than one shows an enrichment of Patronin-RFP at the site of the cut. The graphs in Fig. 7M and S5K show this ratio at the first time point after the cut.

*Quantification of the proportion of centrosomes nucleating microtubules per placode.* EB1 fluorescence intensity was measured on time projections (5–10 time points) in a circular ROI (1.1 μm diameter) around each centrosome in a given placode, and normalised against EB1 fluorescence intensity in the entire placode. This normalised intensity was first measured for centrosomes in a small number of control embryos, of which centrosomes were also manually assessed for microtubule nucleation. This allowed us to manually define a threshold of normalised intensity of 1.15, above which all centrosomes assessed were found to be nucleating microtubules. We then measured EB1 normalised intensity in our entire dataset (231 centrosomes in 9 control embryos and 171 centrosomes in 8 embryos over-expressing γ-tubulin-EGFP) and used the threshold of 1.15 to assess the proportion of centrosomes nucleating microtubules.

**Statistics and reproducibility**. Significance was determined using two-tailed Student's *t*-test, non-parametric Mann–Whitney test for non-Gaussian distribution, unpaired *t*-test with Welch's correction for data with unequal standard deviations, and Kolmogorov–Smirnov (K–S) test for the comparison of cumulative distributions. For the γ-tubulin overexpression experiment (Fig. 3l), the K–S test did not show any significant difference for the cumulative data between the control and γ-tubulin-overexpressing embryos. However, comparisons of cell apical areas in small size windows revealed a significant difference in the distribution of cells with the smallest apical areas (0 μm² < apical area < 5 μm², $p = 0.0361$; 5 μm² < apical area < 10 μm², $p = 0.0361$). Centrosome asymmetries (Fig. 2) were compared using a Kruskal–Wallis test (non-parametric one-way ANOVA) for multiple pairwise comparisons. Results were considered significant when $p < 0.05$. N values and statistical tests used are indicated in the "Results" section as well as in the figure legends.

Figures in most instances display individual cell values, with cell and embryo numbers clearly indicated in the figure and legend. The below table comprehensively lists the statistical analyses based on using individual cell values as well as using average cell values per analysed embryo.

Summary of statistical analyses performed in this study and *p*-values.

| Figure panel | Statistical analysis based on individual cells for all embryos | | | Statistical analysis based on average values of cells for each embryo | | |
|---|---|---|---|---|---|---|
| | Number of cells (number of embryos) | *p*-value | Statistical test | Number of embryos | *p*-value | Statistical test |
| Fig. 4d | Control: 90 (5) ncd::γ-tubulin-EGFP: 90 (5) | <0.0001 | Mann–Whitney test | control: 5 ncd::γ-tubulin-EGFP: 5 | <0.0001 | Unpaired *t*-test |
| Fig. 4g | Control: 125 (5) ncd::γ-tubulin-EGFP: 100 (4) | 0.0003 | Mann–Whitney test | Control: 5 ncd::γ-tubulin-EGFP: 4 | 0.7302 | Mann–Whitney test |
| Fig. 4l | Control: 349 (11) ncd::γ-tubulin-EGFP: 298 (10) | <0.0001 | Mann–Whitney test | Control: 11 ncd::γ-tubulin-EGFP: 10 | 0.0002 | Mann–Whitney test |
| Supplementary Fig. 4e | Control: 480 (12) ncd::γ-tubulin-EGFP: 520 (13) | <0.0001 | Mann–Whitney test | Control: 12 ncd::γ-tubulin-EGFP: 13 | 0.0066 | Unpaired *t*-test |
| Fig. 6g | Control: 440ᵃ (14) Kat80YFP degradFP: 541ᵃ (20) | <0.0001 | Mann–Whitney test | Control: 14 Kat80YFP degradFP: 20 | 0.0086 | Unpaired *t*-test |
| Fig. 6k | Control: 924 (17) Kat80YFP degradFP: 919 (15) | <0.0001 | Mann–Whitney test | Control: 17 Kat80YFP degradFP: 15 | 0.0178 | Unpaired *t*-test |
| Fig. 7h | Control: 955 (13) | <0.0001 | Mann–Whitney test | | 0.0014 | Mann–Whitney test |
| | UAS-Spastin: 999 (19) | | | Control: 13 UAS-Spastin: 19 | | |
| Supplementary Fig. 5c | Placode: 726; epidermis: 726 (15) | <0.0001 | Mann–Whitney test | Placode and epidermis: 15 | 0.0001 | Mann–Whitney test |
| Supplementary Fig. 5i | Control: 600 (12) Kat80YFP degradFP: 595 (11) | <0.0001 | Mann–Whitney test | Control: 12 Kat80YFP degradFP: 11 | 0.0437 | Unpaired *t*-test |
| Fig. 8g | Control: 316 (9) UAS-Patronin-RNAi: 253 (7) | <0.0001 | Mann–Whitney test | Control: 9 UAS-Patronin-RNAi: 7 | 0.0007 | Mann–Whitney test |
| Supplementary Fig. 6c | Control: 250 (5) UAS-Patronin-RNAi: 214 (5) | <0.0001 | Mann–Whitney test | Control: 5 UAS-Patronin-RNAi: 5 | 0.0206 | Unpaired *t*-test |
| Supplementary Fig. 6d | Control: 250 (5) UAS-Patronin-RNAi: 214 (5) | <0.0001 | Mann–Whitney test | Control: 5 UAS-Patronin-RNAi: 5 | 0.0344 | Unpaired *t*-test |
| Supplementary Fig. 6e | Control: 250 (5) UAS-Patronin-RNAi: 214 (5) | 0.3943 | Mann–Whitney test | Control: 5 UAS-Patronin-RNAi: 5 | 0.4417 | Mann–Whitney test |
| Supplementary Fig. 6k | control: 180 (6) UAS-Patronin-RNAi: 150 (5) | <0.0001 | Mann–Whitney test | Control: 6 UAS-Patronin-RNAi: 5 | 0.0221 | Unpaired *t*-test |

ᵃThese numbers correspond to centrosomes, not cells.
Number of independent experiments.

| Figure panel | Number of independent experiments |
|---|---|
| 1b–d | 2 independent experiments |
| 1f, g | 2 independent experiments |
| 2c | 2–3 independent experiments |
| 2d, e | Live: 8 embryos imaged on 2 different days |
| 2f, g | 3 independent experiments |
| 3a, b | 2 independent experiments |
| 3d–d' | 3 independent experiments |
| 3e–e' | 2 independent experiments |
| 3f–f' | 2 independent experiments |
| 4a–d | 3 independent experiments |
| 4f | 4 independent experiments |
| 4g | 2 independent experiments |
| 4h, i | 3 independent experiments |
| 4j–l | 2 independent experiments |
| 5a–a'' | 4 independent experiments |
| 5b | 2 independent experiments |
| 5c | 3 independent experiments |
| 5d–d' | 3 independent experiments |
| 5f | Live: 5 embryos imaged over 3 different days |
| 6b–d | 4 independent experiments |
| 6e–g | 3 independent experiments |
| 6h–h'' | >4 independent experiments |
| 6i–k | 3 independent experiments |
| 7a–c' | >4 independent experiments |
| 7d, e | 3 independent experiments |
| 7f–h | 3 independent experiments |
| 7i–k | Live: embryos imaged over more than 5 different days (numbers of embryos on the figure) |
| 8a–d' | 2 independent experiments |
| 8e–g | 2 independent experiments |
| 8i, j' | 2 independent experiments |
| Suppl.1a–c | 2 independent experiments |
| Suppl.2a–e | 2 independent experiments |
| Suppl.3a–c | Live: 6 embryos imaged over 3 different days |
| Suppl.3e–g | 2 independent experiments |
| Suppl.3h–j | Only 1 staining (5 embryos) |
| Suppl.4a, b | 2 independent experiments |
| Suppl.4c, d | 2 independent experiments |
| Suppl.4e | 2 independent experiments |
| Suppl.4f, g' | 3 independent experiments |
| Suppl.4h, i' | 2 independent experiments |
| Suppl.4j–l'' | Live: embryos imaged over more 5 different days for WT (3 days for gtub OE). Number of embryos on the figure |
| Suppl.5a–c | 3 independent experiments |
| Suppl.5d–f | Live: 6 embryos imaged over 3 different days |
| Suppl.5g–i | 2 independent experiments |
| Suppl.5j–l | Live: embryos imaged over more than 5 different days (numbers of embryos on the figure) |

| | |
|---|---|
| Suppl.6a, b″ | 4 independent experiments |
| Suppl.6f, g | 2 independent experiments |
| Suppl.6h–h″ | 2 independent experiments |
| Suppl.6i–k | 1 experiment |

*p*-values for pairwise comparison in Fig. 2c′

| Dunn's multiple comparisons test | Adjusted *p* value |
|---|---|
| Sas-4-GFP vs. Spd-2-GFP | >0.9999 |
| Sas-4-GFP vs. anti-Asl | >0.9999 |
| Sas-4-GFP vs. Polo-GFP (CRISPR) | <0.0001 |
| Sas-4-GFP vs. anti-γ-tubulin | <0.0001 |
| Sas-4-GFP vs. anti-Cnn | <0.0001 |
| Sas-4-GFP vs. YFP-Centrobin | <0.0001 |
| Spd-2-GFP vs. anti-Asl | >0.9999 |
| Spd-2-GFP vs. Polo-GFP (CRISPR) | <0.0001 |
| Spd-2-GFP vs. anti-γ-tubulin | <0.0001 |
| Spd-2-GFP vs. anti-Cnn | <0.0001 |
| Spd-2-GFP vs. YFP-Centrobin | <0.0001 |
| Anti-Asl vs. Polo-GFP (CRISPR) | <0.0001 |
| Anti-Asl vs. anti-γ-tubulin | <0.0001 |
| Anti-Asl vs. anti-Cnn | <0.0001 |
| Anti-Asl vs. YFP-Centrobin | <0.0001 |
| Polo-GFP (CRISPR) vs. anti-γ-tubulin | >0.9999 |
| Polo-GFP (CRISPR) vs. anti-Cnn | 0.5086 |
| Polo-GFP (CRISPR) vs. YFP-Centrobin | <0.0001 |
| anti-γ-tubulin vs. anti-Cnn | <0.0001 |
| anti-γ-tubulin vs. YFP-Centrobin | <0.0001 |
| Anti-Cnn vs. YFP-Centrobin | <0.0001 |

**Reporting summary**. Further information on research design is available in the Nature Research Reporting Summary linked to this article.

## Data availability
All data generated or analysed during this study are included in this published article (and its supplementary information files).

## Code availibility
Only previously published code and commercially available software packages were used in this publication, as detailed in the "Methods" section. Otracks code[67–69] is available from Dr. Guy Blanchard upon request (gb288@cam.ac.uk).

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

## Acknowledgements
The authors would like to thank the following people; for reagents and fly stocks: Emmanuel Derivery, Debbie Andrew, Jordan Raff, Paul Conduit, Daniel St. Johnston, Claudio Sunkel, Yu-Chiun Wang, Jens Januschke, Cayetano Gonzales, Magali Suzanne, Markus Affolter, Ron Vale; for use of the otracks software: Guy Blanchard; for image analysis: Jérôme Boulanger. The work is supported by the Medical Research Council (file reference number U105178780).

## Author contributions
Conceptualisation, K.R. and G.Gill.; Methodology, K.R., G.Gill. G.Gird.; Investigation, K.R., G.Gill., G.Gird.; Writing-Original Draft, K.R., G.Gill.; Funding Acquisition, K.R.

## Competing interests
The authors declare no competing interests.
