## [Peer Review File · Nature Communications]

REVIEWER COMMENTS

Reviewer #1 (Remarks to the Author):

Gillard et al describe the reorganization and effects of MT arrays during the initial steps of salivary gland internalization in the *Drosophila* embryo. Cells of the gland are known to become post-mitotic before the internalization, and, consistent with this transition, their centrosomal networks are down-regulated, with only the daughter centrosome retaining substantial structure and activity. The microtubule-severing protein Katanin is found to accumulate around the centrosome. The centrosomes are positioned in the apical domain of the epithelial cells, and the experimental elevation of centrosomal MTs or the experimental degradation of Katanin each alter the actomyosin-based apical constriction of the cells. The minus-end binding protein Patronin, known to function with Katanin in other systems, is shown to accumulate in the apical domain, as recently shown during apical constriction in the *Drosophila* ventral furrow, and impacts both MT organization and apical constriction. The paper describes interesting findings and the microscopy is very clear, but a number of issues limit the paper's potential impact.

1. The specific molecular mechanisms examined have been studied in other contexts, and no new insights into specific molecular mechanisms are provided (e.g. how asymmetric centrosome activity arises, how Katanin and Patronin are recruited, how the MT hand off occurs, or how the MTs affect actomyosin). I am not suggesting that all of these mechanisms be investigated in one paper, but I would expect greater mechanistic insight into at least one or two of them for a publication in *Nature Communications*.

2. The manipulations of the molecular mechanisms produce fairly mild effects on gland internalization. The only aspect of gland internalization analyzed is the apical perimeters/areas of cells during the initial stage of the process. The effects on the apical areas is fairly subtle after the manipulations (the gamma-tubulin over-expression, the Katanin degradation, and the Patronin RNAi). The Results section ends with: "Thus, Patronin in the salivary gland placode, once localised to microtubule minus ends within the apical-medial domain, serves to support the organisation of the longitudinal microtubules array that in turn is required for successful apical constriction of cells and formation of the wild-type tubular organ." Elsewhere in the Results the authors also suggest that the mechanisms described in this paper would affect formation of the tubular organ. However, the full effects of the perturbations on the internalization of the organ are not described. Since the effects on the initial apical constriction are fairly subtle, the overall internalization might occur fairly normally. It may be that studied mechanisms are not that important for the internalization. Substantial new analyses would be needed to argue otherwise.

3. Large numbers of data points are shown in graphs. They represent values from large numbers of cells analyzed from a smaller number of embryos. It was unclear what the N values for the statistical tests represent: the total number of cells or the total number of embryos (e.g. "n=284 cells from 8 embryos; statistical significance was deduced by Mann-Whitney test of comparison as $p < 0.0001$ ")? Since multiple cells from one embryo are not independent of one another, the N values for statistical tests should be based on the number of embryos (with the values from the multiple cells of an individual embryo somehow collapsed into one representative value for the embryo). It would also be good to show these smaller numbers of embryo values in graphs (perhaps in supplemental data). I worry that subtle effects are shown to be highly statistically significant because of artificial inflation of the sample sizes (see "What exactly is 'N' in cell culture and animal experiments" PMID

29617358).

Reviewer #2 (Remarks to the Author):

Non-centrosomal microtubules are likely the normal mode of microtubule patterning in most differentiated cell types, yet how these arrays are established across cell types remains largely unknown. Several models have been proposed, including direct nucleation from non-centrosomal sites and release of microtubules from the centrosome and their subsequent capture at non-centrosomal sites and these mechanisms likely vary by cell type. Here, Gillard et al., explore the formation of a non-centrosomal microtubule arrays in cells of the developing *Drosophila* salivary glands. They explore the MTOC function at the centrosome upon mitotic exit, finding an asymmetry in the localization of various PCM proteins between the two centrosomes including CNN, g-tubulin, and Polo and some evidence for asymmetric MT nucleation as revealed by EB1 imaging. Overexpression of g-tubulin in these cells leads to defects in tubulin localization and to wavy cell junctions. They find evidence for a role for katanin in shaping the microtubule network in these cells as the p80 subunit localizes near centrosomes on the apical surface and p80 degradation change tubulin localization. Finally, Patronin also appears to play a role in shaping the microtubule network as Patronin relocates from junction to minus ends of microtubules consistent with its known role as a minus end binding protein and this relocalization depends on microtubules and p80. A modest removal of Patronin from these cells also leads to changes in the microtubule network and issues with apical constriction, consistent with a role for microtubules in apical constriction as the authors have previously reported. My largest issue and the proposed novelty presented here is the lack of data supporting a release and capture model. Some of the data perhaps hints at this, but the current data seem very far from demonstrating this mechanism. These and other comments are listed below:

Major comments:

1) Need better documentation of non-centrosomal MTs: One of the major conclusions of this paper (building from previous work by these authors) is the appearance of non-centrosomal microtubules in this cell type. The paper should therefore start with a presentation of these microtubules and their orientation. We do not see microtubule in these cells and their orientation until Fig. 3N and it is not until Figure 6 that we get a good look at the non-centrosomal MTs running perpendicular to the apical surface. Fig 3D-E and S3A would benefit from zoomed in insets to see MTs and co-labeling to see where centrosomes are.

2) Need better documentation of centrosomal MTOC function in time and space: In Figure 2, the authors argue that MTs are emanating from one of the two centrosomes, but when is this relative to the generation of non-centrosomal MTs? In addition, as presented, EB1 data is unconvincing: time projection is hard to see and interpret. Need a legend for color and also an inset with a blow up. I would rather see this analysis as EB1 comets emanating from Cnn positive vs. Cnn negative centrosomes.

3) Issues with g-tubulin overexpression: It is strange that g-tubulin overexpression has this effect, suggesting that g-tubulin is limiting over all other members of the g-TuRC. In most cell types, there is a ton of soluble g-tubulin, so it is surprising that this overexpression would have any effect. I worry

that this effect could be a dominant negative effect of overexpressing g-tubulin with a tag (on the C-terminus) Could the authors demonstrate that this transgene is functional? Otherwise, it is hard to interpret phenotypes.

p.7: The authors state that “Excessive amounts of microtubules generated in placodal cells of *ncd::g-tubulin-EGFP* embryos, though nucleated at centrosomes, in the placodal cells appeared to contribute to an enlarged non-centrosomal array (Fig. 3 N, O)” What is the basis for this statement? The authors show no evidence that microtubules are nucleated at centrosomes in this *ncd::g-tubulin-EGFP* expressing strain. To make this statement, the authors would need to show increased centrosomal nucleation at centrosomes and that non-centrosomal MTs originate from the centrosome. I see neither of these pieces of data presented. Currently this statement rests on the fact that following expression of *ncd::g-tubulin-EGFP*, g-tubulin-EGFP localizes to both centrosomes. In many systems, including *Drosophila*, the g-tubulin is dispensable for MT nucleation (Rogers and Rogers, 2008; Sallee et al., 2018).

4) no direct evidence for release and capture mechanism: The title of this paper is “A release and capture mechanism generates an essential non-centrosomal microtubule array...”, however, the authors show no direct evidence for this. As I see it, the authors show that at some point one of two centrosomes might remain active as an MTOC (see above), then microtubule become non-centrosomal. You can perturb microtubule localization in various ways and they become localized parallel to the apical surface which is quite small and where centrosomes also localize. To show release and capture, you would need to either create marks on microtubules to follow them over time or at very least in the various perturbations the authors employ (i.e. Figure 5E) verify that the centrosome is still indeed functioning as an MTOC, i.e. through EB1 tracking, and/or microtubule regrowth.

Minor comments:

You can't know that the asymmetrically activated centrosome in placodal cells is the daughter centrosome without pulse-chase labeling as was done in neuroblasts.

Figures and panels are generally referred to out of order: Fig 2A is cited first, then 1a, 1b, 1f, 1g

2C: scale here makes it hard to know the asymmetry index for things closer to one

Katanin localization in Figure 4D is unconvincing

Need to verify that Spastin expression removes MTs, not just acetylated MTs

Figure 7—need cross section of MT phenotypes

Fig 7C, D need genotype labels on Figure

Response to Reviewers' Comments

We are pleased the reviewers found our data and manuscript 'interesting findings' and the 'microscopy [to be] very clear'. We would like to thank the reviewers for their comments, which have led to various additions and changes to the manuscript that we feel have made it even stronger and more conclusive. In particular, we have investigated the release-and-capture mechanism in more detail. We respond to all comments and issues raised by the reviewers in the point-by-point rebuttal below.

Reviewer #1 (Remarks to the Author):

Gillard et al describe the reorganization and effects of MT arrays during the initial steps of salivary gland internalization in the *Drosophila* embryo. Cells of the gland are known to become post-mitotic before the internalization, and, consistent with this transition, their centrosomal networks are down-regulated, with only the daughter centrosome retaining substantial structure and activity. The microtubule-severing protein Katanin is found to accumulate around the centrosome. The centrosomes are positioned in the apical domain of the epithelial cells, and the experimental elevation of centrosomal MTs or the experimental degradation of Katanin each alter the actomyosin-based apical constriction of the cells. The minus-end binding protein Patronin, known to function with Katanin in other systems, is shown to accumulate in the apical domain, as recently shown during apical constriction in the *Drosophila* ventral furrow, and impacts both MT organization and apical constriction. The paper describes interesting findings and the microscopy is very clear, but a number of issues limit the paper's potential impact.

1. The specific molecular mechanisms examined have been studied in other contexts, and no new insights into specific molecular mechanisms are provided (e.g. how asymmetric centrosome activity arises, how Katanin and Patronin are recruited, how the MT hand off occurs, or how the MTs affect actomyosin). I am not suggesting that all of these mechanisms be investigated in one paper, but I would expect greater mechanistic insight into at least one or two of them for a publication in *Nature Communications*.

With regards to the suggested lack of mechanism or detail on Patronin recruitment, we demonstrated already in the original submission that the apical-medial recruitment of Patronin requires an intact MT network (Fig. 6) and also Katanin (Fig. S5), suggesting that microtubule severing is required for Patronin relocalisation from the cell cortex or cytoplasm to the minus-end of microtubules. To directly confirm that Patronin is recruited to, and captured by, microtubule minus-ends in placodal cells in vivo, we now performed laser-ablation experiments (in comparison to appropriate controls) both in early placodes (early stage 11), where microtubules emanate from centrosomes and are localised within the apical domain, and in later placodes (stage 12), where microtubules are non-centrosomal and oriented along the apical-basal axis of the cells in a longitudinal manner. In both situations, microtubule ablation induced the very rapid and specific recruitment of Patronin to severed microtubules, confirming that generation of free minus-ends is sufficient to induce Patronin capture by microtubules. This is now shown in Figure 6I-M and Supplemental Figure S5J-L and Supplemental Movies 2-5.

*In addition, we previously showed that the spectraplakin Short Stop (Shot) is important to promote apical-medial actomyosin recruitment, probably by linking MTs to actin through its EF hand and Gas2 domain, and CH domains, respectively (Booth, A. J. R., Blanchard, G. B., Adams, R. J., & Röper, K. (2014). A Dynamic Microtubule Cytoskeleton Directs Medial Actomyosin Function during Tube Formation. *Developmental Cell*, 29(5), 562–576. <http://doi.org/10.1016/j.devcel.2014.03.023>). We have now visualised Shot localisation in Patronin-depleted embryos where microtubules fail to reorganise into the longitudinal array*

in placodal cells. Patronin-depletion led to a defect in the apical-medial recruitment of Shot, thus explaining the loss of apical-medial actomyosin when microtubule reorganisation is prevented by reducing Patronin.

2. The manipulations of the molecular mechanisms produce fairly mild effects on gland internalization. The only aspect of gland internalization analysed is the apical perimeters/areas of cells during the initial stage of the process. The effects on the apical areas is fairly subtle after the manipulations (the gamma-tubulin over-expression, the Katanin degradation, and the Patronin RNAi). The Results section ends with: “Thus, Patronin in the salivary gland placode, once localised to microtubule minus ends within the apical-medial domain, serves to support the organisation of the longitudinal microtubules array that in turn is required for successful apical constriction of cells and formation of the wild-type tubular organ.” Elsewhere in the Results the authors also suggest that the mechanisms described in this paper would affect formation of the tubular organ. However, the full effects of the perturbations on the internalization of the organ are not described. Since the effects on the initial apical constriction are fairly subtle, the overall internalization might occur fairly normally. It may be that studied mechanisms are not that important for the internalization. Substantial new analyses would be needed to argue otherwise.

*We apologise if our introduction did not put this into context clearly enough. We published previously (Booth, A. J. R., Blanchard, G. B., Adams, R. J., & Röper, K. (2014). A Dynamic Microtubule Cytoskeleton Directs Medial Actomyosin Function during Tube Formation. *Developmental Cell*, 29(5), 562–576. <http://doi.org/10.1016/j.devcel.2014.03.023>) how the non-centrosomal microtubule array in the placodal cells that are undergoing morphogenesis to form the tubes of the salivary glands is important for apical-medial actomyosin pulsatile activity. This activity in turn is key to drive the apical constriction and cell wedging that drives the tube morphogenesis (Sánchez-Corrales, Y. E., Blanchard, G. B., & Röper, K. (2018)). Radially patterned cell behaviours during tube budding from an epithelium. *eLife*, 7. <http://doi.org/10.7554/eLife.35717>). Thus, affecting apical constriction in the placodal cells will always negatively affect tube morphogenesis.*

*But the reviewer is correct that the perturbations do not lead to the most severe effects of preventing tube invagination completely. This is not surprising, though, as most morphogenetic processes operate with many belts-and-braces in place (such as mesoderm invagination in *Drosophila*, where many mutants only show delays or loss of synchronous invagination, and only the abolition of the most upstream transcription factors *Twist* and *Snail* will actually abolish the process completely). Similarly, during the formation of the salivary gland placode many mutations even in known and published key factors will lead to the formation of aberrant tubes (in shape, size or position), but most, apart from mutants in the most upstream tissue-specifying transcription factors, do not abolish invagination (Xu, N., Keung, B., & Myat, M. M. (2008). *Rho GTPase controls invagination and cohesive migration of the Drosophila salivary gland through Crumbs and Rho-kinase. Developmental Biology*, 321(1), 88–100. <http://doi.org/10.1016/j.ydbio.2008.06.007>; Maybeck, V., & Röper, K. (2009). A targeted gain-of-function screen identifies genes affecting salivary gland morphogenesis/tubulogenesis in *Drosophila*. *Genetics*, 181(2), 543–565. <http://doi.org/10.1534/genetics.108.094052>; Kolesnikov, T., & Beckendorf, S. K. (2007). *18 wheeler regulates apical constriction of salivary gland cells via the Rho-GTPase-signaling pathway. Developmental Biology*, 307(1), 53–61. <http://doi.org/10.1016/j.ydbio.2007.04.014>).*

*We previously published late stage gland phenotypes for the situation where microtubules are depleted using Spastin overexpression (Booth, A. J. R., Blanchard, G. B., Adams, R. J., & Röper, K. (2014). A Dynamic Microtubule Cytoskeleton Directs Medial Actomyosin Function during Tube Formation. *Developmental Cell*, 29(5), 562–576. <http://doi.org/10.1016/j.devcel.2014.03.023>). This is copied from Supplemental Figure S2 of the above paper:*

We are very happy to provide similar late stage images, and include those now as Supplemental Figure S7 and also copied below:

3. Large numbers of data points are shown in graphs. They represent values from large numbers of cells analyzed from a smaller number of embryos. It was unclear what the N

values for the statistical tests represent: the total number of cells or the total number of embryos (e.g. “n=284 cells from 8 embryos; statistical significance was deduced by Mann-Whitney test of comparison as $p < 0.0001$ ”)? Since multiple cells from one embryo are not independent of one another, the N values for statistical tests should be based on the number of embryos (with the values from the multiple cells of an individual embryo somehow collapsed into one representative value for the embryo). It would also be good to show these smaller numbers of embryo values in graphs (perhaps in supplemental data). I worry that subtle effects are shown to be highly statistically significant because of artificial inflation of the sample sizes (see “What exactly is ‘N’ in cell culture and animal experiments” PMID 29617358).

We are very happy to explain the use of statistics in the manuscript, as requested by this reviewer. We have followed standard practice of many labs in many studies of similar types, stating embryo and cell number analysed, and basing statistics, usually Mann-Whitney tests, on cell numbers. These types of statistics have been used by us and others studying and quantifying aspects of morphogenetic processes many times in the past and successfully published. But we are happy to provide clarity and detail. Examples of recent papers using the same type of statistical analyses are listed here:

- Dehapiot, B., Clément, R., Alégot, H., Gaszón-Gerhát, G., Philippe, J.-M., & Lecuit, T. (2020). Assembly of a persistent apical actin network by the formin Frl/Fmnl tunes epithelial cell deformability. *Nature Cell Biology*, 1–21. <http://doi.org/10.1038/s41556-020-0524-x>

- Finegan TM, Hervieux N, Nestor-Bergmann A, Fletcher AG, Blanchard GB, Sanson B. (2019) The tricellular vertex-specific adhesion molecule Sidekick facilitates polarised cell intercalation during *Drosophila* axis extension. *PLoS Biol*;17(12):e3000522. doi: 10.1371/journal.pbio.3000522. PMID: 31805038; PMCID: PMC6894751.

- Ko, C. S., Tserunyan, V., & Martin, A. C. (2019). Microtubules promote intercellular contractile force transmission during tissue folding. *Journal of Cell Biology*, 218(8), 2726–2742. <http://doi.org/10.1083/jcb.201902011>

- Uechi, H., & Kuranaga, E. (2019). The Tricellular Junction Protein Sidekick Regulates Vertex Dynamics to Promote Bicellular Junction Extension. *Developmental Cell*, 50(3), 327–338.e5. <http://doi.org/10.1016/j.devcel.2019.06.017>

- Clément, R., Dehapiot, B., Collinet, C., Lecuit, T., & Lenne, P.-F. (2017). Viscoelastic Dissipation Stabilizes Cell Shape Changes during Tissue Morphogenesis, 1–16. <http://doi.org/10.1016/j.cub.2017.09.005>

We would also like to explain why representing cell data points in our view provides a better understanding and truer representation of the effects that we induce genetically: The tissue-specific tools such as the UAS-Gal4 system we use to overexpress or degrade components do not behave identical in all cells due to the genetics employed. Transgenes tend to not be expressed at identical levels from cell to cell across the placode. Thus, although we agree that cells within one placode are not independent from one another, they are also not identical in their genetic background, their expression level of transgenes and thus their response to such modifications.

Nonetheless, we have also performed statistics not only on the cell number but also by averaging cell data for each individual embryo and basing the statistics instead on average values per embryo, and all results are significant. This is now added in the Material and Methods section as a table, but we prefer to maintain the graphical representation of cells in the figure for the reasons set out above. For clarity, we have added the cell and embryo numbers below each graph (e.g. ‘(542c/15e)’ indicating 542 cells from 15 embryos).

Reviewer #2 (Remarks to the Author):

Non-centrosomal microtubules are likely the normal mode of microtubule patterning in most differentiated cell types, yet how these arrays are established across cell types remains largely unknown. Several models have been proposed, including direct nucleation from non-centrosomal sites and release of microtubules from the centrosome and their subsequent capture at non-centrosomal sites and these mechanisms likely vary by cell type. Here, Gillard et al., explore the formation of a non-centrosomal microtubule arrays in cells of the developing *Drosophila* salivary glands. They explore the MTOC function at the centrosome upon mitotic exit, finding an asymmetry in the localization of various PCM proteins between the two centrosomes including CNN, γ -tubulin, and Polo and some evidence for asymmetric MT nucleation as revealed by EB1 imaging. Overexpression of γ -tubulin in these cells leads to defects in tubulin localization and to wavy cell junctions. They find evidence for a role for katanin in shaping the microtubule network in these cells as the p80 subunit localizes near centrosomes on the apical surface and p80 degradation change tubulin localization. Finally, Patronin also appears to play a role in shaping the microtubule network as Patronin relocates from junction to minus ends of microtubules consistent with its known role as a minus end binding protein and this relocalisation depends on microtubules and p80. A modest removal of Patronin from these cells also leads to changes in the microtubule network and issues with apical constriction, consistent with a role for microtubules in apical constriction as the authors have previously reported. My largest issue and the proposed novelty presented here is the lack of data supporting a release and capture model. Some of the data perhaps hints at this, but the current data seem very far from demonstrating this mechanism. These and other comments are listed below:

Major comments:

1) Need better documentation of non-centrosomal MTs: One of the major conclusions of this paper (building from previous work by these authors) is the appearance of non-centrosomal microtubules in this cell type. The paper should therefore start with a presentation of these microtubules and their orientation. We do not see microtubule in these cells and their orientation until Fig. 3N and it is not until Figure 6 that we get a good look at the non-centrosomal MTs running perpendicular to the apical surface. Fig 3D-E and S3A would benefit from zoomed in insets to see MTs and co-labeling to see where centrosomes are.

The detailed documentation of the non-centrosomal microtubule array has been performed and published by us in 2014 in Developmental Cell (Booth, A. J. R., Blanchard, G. B., Adams, R. J., & Röper, K. (2014). A Dynamic Microtubule Cytoskeleton Directs Medial Actomyosin Function during Tube Formation. Developmental Cell, 29(5),562–576. <http://doi.org/10.1016/j.devcel.2014.03.023>), and we extensively comment and cite our previous study. We now include as Supplemental Figure S1 confocal images of the apical domain of placodal cells to show the microtubule rearrangement occurring. In addition, we have included a schematic illustrating the rearrangement as Figure 1 A”.

2) Need better documentation of centrosomal MTOC function in time and space: In Figure 2, the authors argue that MTs are emanating from one of the two centrosomes, but when is this relative to the generation of non-centrosomal MTs?

As published previously in Booth et al. 2014 and also in Sanchez-Corrales et al., 2018, the morphogenetic changes in the salivary gland placode during initial tube formation do not occur homogeneously across the whole placode. Apical constriction driven by apical-medial actomyosin starts at the position of the future invagination pit at late stage 10/early stage 11, and with cells constricting and tissue bending commencing, cells begin to internalise. New coronae of cells near the invagination pit are the next to apically constrict, thus the area of

apical constriction and of increased apical-medial actomyosin is expanding across the placode, starting from the position of the forming pit. The change in microtubule orientation and formation of the longitudinal non-centrosomal array mirrors the other changes and initiates at the position of the future pit and spreads radially across the placode primordium from here. We aimed to illustrate this in Figure 1A'-A''' and in referring to our previously published studies, but realise that this required more description and have now added further explanation to the introduction. Again, this was shown and quantified in our previous published work (Booth, A. J. R., Blanchard, G. B., Adams, R. J., & Röper, K. (2014). A Dynamic Microtubule Cytoskeleton Directs Medial Actomyosin Function during Tube Formation. *Developmental Cell*, 29(5), 562–576. <http://doi.org/10.1016/j.devcel.2014.03.023>), as illustrated in the figures below reproduced from this publication.

Figure 2D (Booth et al., 2014):

Supplemental Figure S1A-B'', (Booth et al. 2014):

In addition, as presented, EB1 data is unconvincing: time projection is hard to see and interpret. Need a legend for color and also an inset with a blow up. I would rather see this analysis as EB1 comets emanating from Cnn positive vs. Cnn negative centrosomes.

We apologise that the images were not clear enough. We have changed panel Fig. 2D to now show a projection of 30 time frames of a EB1-GFP Cnn-RFP movie that is now shown as Supplemental Movie 1, as this very clearly illustrates the EB1 comets.

It is unfortunately technically currently not possible to undertake a live-analysis of Cnn-positive versus Cnn-negative centrosomes showing EB1 comets emanating or not, respectively. All available and functioning live labels of EB1, Cnn and other centrosomal

components that would be required to be combined for such analysis are either tagged with GFP or mCherry, hence prohibiting triple colour analyses for now.

3) Issues with g-tubulin overexpression: It is strange that g-tubulin overexpression has this effect, suggesting that g-tubulin is limiting over all other members of the g-TuRC. In most cell types, there is a ton of soluble g-tubulin, so it is surprising that this overexpression would have any effect. I worry that this effect could be a dominant negative effect of overexpressing g-tubulin with a tag (on the C-terminus) Could the authors demonstrate that this transgene is functional? Otherwise, it is hard to interpret phenotypes.

p.7: The authors state that “Excessive amounts of microtubules generated in placodal cells of *ncd::g-tubulin-EGFP* embryos, though nucleated at centrosomes, in the placodal cells appeared to contribute to an enlarged non-centrosomal array (Fig. 3 N, O)” What is the basis for this statement? The authors show no evidence that microtubules are nucleated at centrosomes in this *ncd::g-tubulin-EGFP* expressing strain. To make this statement, the authors would need to show increased centrosomal nucleation at centrosomes and that non-centrosomal MTs originate from the centrosome. I see neither of these pieces of data presented. Currently this statement rests on the fact that following expression of *ncd::g-tubulin-EGFP*, g-tubulin-EGFP localizes to both centrosomes. In many systems, including *Drosophila*, the g-tubulin is dispensible for MT nucleation (Rogers and Rogers, 2008; Sallee et al., 2018).

*We agree that would be interesting to understand how the γ -tubulin overexpression, using the *ncd::g-tubulin-EGFP* transgene, works in the placodal cells. Ultimately, though, it will always be an artificial overexpression situation, and what we are really interested in and are analysing are the overexpression effects, and these are telling: placodal cells show an increase in microtubules that we quantitatively document, now adding evidence showing that these microtubules are non-centrosomal based on the increase in the apical-medial accumulation of Patronin (as quantified by its medial/cortical ratio, see Fig. 3P-R) and likely result from the increased centrosomal nucleation we observe (Fig. S4H-J). Concomitant with this non-centrosomal microtubule increase we see an enlarged apical-medial actomyosin pool and increased apical constriction, again quantified.*

*We agree that in the original version of the manuscript the statement cited above (‘Excessive amounts of microtubules generated in placodal cells of *ncd::g-tubulin-EGFP* embryos, though nucleated at centrosomes, in the placodal cells appeared to contribute to an enlarged non-centrosomal array’) was based on the correlation of increased γ -tubulin labeling at both centrosomes together with the presence of an enlarged and denser non-centrosomal microtubule array (as still shown in the revised version in Fig. 3 N-R). Our immunostaining data using an antibody against γ -tubulin in the *g-tubulin::EGFP* strain did not detect γ -tubulin staining or localisation anywhere else but at centrosomes, strongly suggesting that **de novo** nucleation of microtubules in placodal cells can only take place at centrosomes.*

*We now show time-lapse imaging using EB1-mCherry *ncd::g-tubulin-EGFP* and quantification of centrosomal nucleation to provide evidence of increased nucleation at centrosomes upon γ -tubulin overexpression (Fig. S4H-J), that is concomitant with an enlarged non-centrosomal array as visualised by increased apical-medial Patronin (Fig.3 P-R and Fig. S4F-G”), supporting that the increase in nucleation drives the increase in microtubule array.*

*We would like to disagree, though, with the reviewer’s statement and citations that ‘In many systems, including *Drosophila*, the γ -tubulin is dispensible for MT nucleation (Rogers and Rogers, 2008; Sallee et al., 2018).’ Of these references, the first only describes depletion of centrosomes in S2 tissue culture cells, but not in *Drosophila* tissues, and thus does not support the statement made. In fact, the authors of the study themselves state in the*

*abstract: 'Furthermore, steady-state interphase microtubule levels are not changed by codepleting both γ -tubulins. However, γ -tubulin RNAi delays microtubule regrowth after depolymerization, suggesting that it may function partially redundantly with another pathway.' Which seems to rather indicate some effect on nucleation ability even in S2 cells in vitro. The second cited paper looks at *C. elegans* intestinal cells only, not a variety of tissues or organisms, and even in this situation, the study's authors show there is a subset of microtubules that are sensitive to γ -tubulin removal, just not all microtubules.*

*Coming back to our argument above, we came across the *ncd:: γ -tubulin* transgene as a tool to increase microtubule nucleation (as shown in Fig. S4) and enhance the non-centrosomal array (as shown in Fig. 3 N-R). Thus, how important wild-type levels of γ -tubulin are for all microtubule nucleation in the salivary gland placodal cells in the wild-type is not the point we aim to make here.*

*All we want to show and have documented is that during the generation of the non-centrosomal array, **de novo** nucleation still occurs at one centrosome and nowhere else in the cells, as we can detect γ -tubulin only at centrosomes. Thus, we exclude a mechanism of relocalisation of microtubule nucleators as the way of generating this array, as opposed to what has been described in other systems (Brodu, V., Baffet, A. D., Le Droguen, P.-M., Casanova, J. & Guichet, A. A Developmentally Regulated Two-Step Process Generates a Noncentrosomal Microtubule Network in *Drosophila* Tracheal Cells. *Developmental Cell* **18**, 790–801 (2010); Feldman, J. L. & Priess, J. R. A Role for the Centrosome and PAR-3 in the Hand-Off of MTOC Function during Epithelial Polarization. *Current Biology* **22**, 575–582 (2012).).*

*With regards to the functionality of the transgene or its ability to rescue a mutant: The transgene used, *w1118; P{ncd- γ Tub37C.GFP}F13F3*, does not express γ Tub37C under its own promoter, but under the *ncd* promoter that is ubiquitously expressed during embryogenesis. Rescue of mutants with transgenes that are not under their endogenous promoter is notoriously difficult and usually impossible, as in most cases the precise timing, tissue distribution and level of expression is key to the function of a protein. Thus, we would not expect this transgene to rescue the null mutant, and although we have not attempted ourselves, we have data communicated from colleagues that confirm this (i.e. the transgene does not rescue the mutant).*

*But, for our stated purpose of using this transgene as a tool to increase *de novo* microtubule nucleation at centrosomes, the transgene appears to function correctly: (1) it localises to centrosomes only, as the antibody demonstrates for endogenous γ -tubulin, (2) as quantified by us and now shown in Fig.S4 it leads to an increase in microtubule nucleation with nucleation at both centrosomes.*

4) no direct evidence for release and capture mechanism: The title of this paper is “A release and capture mechanism generates an essential non-centrosomal microtubule array...”, however, the authors show no direct evidence for this. As I see it, the authors show that at some point one of two centrosomes might remain active as an MTOC (see above), then microtubule become non-centrosomal. You can perturb microtubule localization in various ways and they become localized parallel to the apical surface which is quite small and where centrosomes also localize. To show release and capture, you would need to either create marks on microtubules to follow them over time or at very least in the various perturbations the authors employ (i.e. Figure 5E) verify that the centrosome is still indeed functioning as an MTOC, i.e. through EB1 tracking, and/or microtubule regrowth.

As the reviewer themselves state, we show plenty of corroborating evidence for the individual parts of the proposed release-and-capture mechanism. We would posit that this is how such models or mechanisms are experimentally addressed in most studies. In addition, the visualisation of the release and capture mechanism requires high resolution (both spatial and temporal) imaging of the placode or photo-manipulation using photoactivable or

photoconvertible strains of tubulin. These are all very challenging experiments in such tiny cells (apical area of max 2-3 μm in the cells near the invagination pit). However, we strongly believe that our previous data combined with our additional successful experiments strongly support that a release and capture mechanism is at play:

To recapitulate our previous and newly added data:

- 1) we already showed and published the change from centrosomal to non-centrosomal microtubule array (Booth et al. 2014).*
- 2) We now show nucleation from a single centrosome per cell, with Katanin localising to MT ends at this centrosome at the time that the change in the MT array happens.*
- 3) We find no evidence of relocalisation of MT nucleators away from centrosomes (in particular γ -tubulin), demonstrating that de novo nucleation can only occur at centrosomes.*
- 4) To visualise release of microtubules generated at centrosomes, we have performed experiments using a photoconvertible form of tubulin, but have not had success with any of our imaging systems to in vivo in live embryos focus the conversion on a small enough region to be able to convert individual or small groups of microtubule ends near centrosomes. Instead, we have live-imaged centrosomes labeled using As1-mCherry and microtubules labeled using Jupiter-GFP in a single apical section at the level of the nucleating centrosome. In these sections, we can frequently detect bright Jupiter-GFP foci moving away and losing contact with centrosomes. As Jupiter will label entire microtubules, we interpret these foci moving away from centrosomes as microtubule ends being released.*
- 5) We show that microtubule minus-ends away from centrosomes colocalise with Patronin, and Patronin localisation depends on the MT minus-ends and Katanin function. Additionally, we now also demonstrate that the generation of free microtubule minus-ends via laser ablation within the apical region of placodal cells is sufficient to induce Patronin recruitment to these microtubule ends. This underlines the capture part of the mechanism, demonstrating that once free minus-ends are generated, usually by Katanin severing, or experimentally by laser-induced severing, these are quickly bound by Patronin.*
- 6) Finally, we show that both Katanin and Patronin depletion prevents proper microtubule reorganisation.*
- 7) Also, the fact that microtubule depletion, Katanin-degradation as well as Patronin knock-down in the placode all lead to similar problems in apical constriction and thereby orderly and wild-type invagination of the tubular organ (as further demonstrated by late phenotypes in Fig. S7), strongly suggests them working in the same pathway.*

The reviewer discusses above for us 'in the various perturbations the authors employ to verify that the centrosome is still indeed functioning as an MTOC, i.e. through EB1 tracking, and/or microtubule regrowth.'

This would be feasible technically in some though not all of the conditions, though we are unsure how this would prove the 'release-and-capture' any further than what we already show. It would confirm that any perturbation would not inadvertently abolish nucleation at the centrosome (though it is unclear to us why Katanin- or Patronin-depletion would have such effect), but we are unsure what it shows beyond that point. Also, the fact that there is a centrosome-focused apical, rather than a longitudinal, array under Katanin-degradation and Patronin-depletion in our view already demonstrates that overall nucleation is not affected.

So beyond demonstrating each step of the proposed release-and-capture process, including further live imaging and live-perturbation as outlined above, we do not know how else to verify the steps of the process in vivo.

Minor comments:

You can't know that the asymmetrically activated centrosome in placodal cells is the daughter centrosome without pulse-chase labeling as was done in neuroblasts.

*We are happy to phrase this more cautiously, to say data suggest it to be the daughter centriole due to Centrobilin-labelling. Centrobilin-labeling, in analogy to the situation in Drosophila larval neuroblasts (Januschke, J., Llamazares, S., Reina, J., & Gonzalez, C. (2011). <http://doi.org/10.1038/ncomms1245>; Januschke, J., Reina, J., Llamazares, S., Bertran, T., Rossi, F., Roig, J., & González, C. (2013). Centrobilin controls mother-daughter centriole asymmetry in Drosophila neuroblasts. *Nature Cell Biology*, 15(3), 241–248. <http://doi.org/10.1038/ncb2671>) would be highly consistent with this being the daughter centriole.*

We now say on page 4 and 7:

Page 4: 'Here, we describe our discovery of a step-wise process that implements these changes in the salivary gland placode: as part of concluding embryonic mitoses, the cells of the placode are the first to enter a G1 phase with concomitant loss of microtubule nucleation capacity of the Centrobilin-enriched centrosome, a loss that we show is important to ensure proper morphogenesis.'

Page 7: 'Taken together these result show that only a single centrosome in the secretory cells of the salivary gland placode during early tube morphogenesis retains microtubule nucleation capacity. Similar to fly neuroblasts, Centrobilin-labelling suggests that this could be the daughter centriole.'

Figures and panels are generally referred to out of order: Fig 2A is cited first, then 1a, 1b, 1f, 1g

Thanks for alerting us to this. We have adjusted this and made sure figure panels are referenced in order.

2C: scale here makes it hard to know the asymmetry index for things closer to one

Because of this issue we had added Figure 2C' in the original submission, which is a magnification of the plot shown in Figure 2C around the value of 1. We felt it would be important to show both more extreme data points such as in 2C, but also home in on the symmetrical distribution around 1 in Figure 2C'. We have now explained this clearer in the figure legend.

Katanin localization in Figure 4D is unconvincing

We agree and have changed this to a more representative and clearer image.

Need to verify that Spastin expression removes MTs, not just acetylated MTs

We know that Spastin efficiently removes microtubules and not just acetylated microtubules. This has been shown in previous studies by other labs using Spastin expression as a means of removing microtubules including:

- Brodu, V., Baffet, A. D., Le Droguen, P.-M., Casanova, J., & Guichet, A. (2010). A developmentally regulated two-step process generates a noncentrosomal microtubule network in *Drosophila* tracheal cells. *Developmental Cell*, 18(5), 790–801. <http://doi.org/10.1016/j.devcel.2010.03.015>
- Corrigan, D., Walther, R. F., Rodriguez, L., Fichelson, P., & Pichaud, F. (2007). Hedgehog signaling is a principal inducer of Myosin-II-driven cell ingression in *Drosophila* epithelia. *Developmental Cell*, 13(5), 730–742. <http://doi.org/10.1016/j.devcel.2007.09.015>
- Bulgakova, N. A., Grigoriev, I., Yap, A. S., Akhmanova, A., & Brown, N. H. (2013). Dynamic microtubules produce an asymmetric E-cadherin-Bazooka complex to maintain segment boundaries. *The Journal of Cell Biology*, 201(6), 887–901. <http://doi.org/10.1083/jcb.201211159>

We also include a Figure for the reviewer to demonstrate loss of DM1A (anti α -tubulin antibody) labeling and YL1/2 (anti tyrosinated α -tubulin antibody) labeling in the salivary gland placode below:

Figure 1 for Reviewer 2.
UAS-Spastin x fkhGal4-driven microtubule depletion

Figure 7—need cross section of MT phenotypes
Fig 7C, D need genotype labels on Figure

We have rearranged part of this Figure, so it now includes cross sections of the magnifications, and these are also placed in a position where the genotype is obvious.

REVIEWER COMMENTS

Reviewer #1 (Remarks to the Author):

The authors have effectively addressed my past concerns, providing substantial additions of data addressing mechanisms of molecular interactions, as well as providing broader developmental context and clearer explanation of statistical tests. Overall, the paper should be of board interest. It dissects major steps of non-centrosomal microtubule array formation in epithelial cells, and shows that this microtubule reorganization promotes actomyosin-based apical constriction and associated tissue morphogenesis *in vivo*. The approaches are elegant, the data are clear, and the paper is very well-written.

Reviewer #2 (Remarks to the Author):

Here Gillard et al. present their revised manuscript, adding significant and important experiments that further bolster their claim that non-centrosomal microtubules in epithelial cells of the *Drosophila* salivary gland are captured after release from the centrosome. Release and capture has been proposed for many years as a mechanism to generate non-centrosomal microtubule arrays, however little direct evidence exists, especially *in vivo*, to support this mechanism. The authors now add additional data, including following Jupiter:GFP as it moves away from centrosomes (Fig. 4E, F) and the ability of Patronin to localize to free microtubule minus ends that have been created by severing. These experiments strengthen the authors' claim, however, I think that it is still open to interpretation whether they have demonstrated release and capture. As they say in their rebuttal, they still present plenty of corroborating evidence, however correlation does not equal causation. Ultimately, it will be up to the reader to decide, but I think the authors could significantly strengthen their paper with the following revisions:

Major Comments:

Improve Imaging: While I feel reasonable confident that the authors have found several conditions to affect the centrosomal microtubule array, I am still left unconvinced by the phenotypes surrounding the non-centrosomal microtubule arrays. The issue arises from the fact that the authors show apical and therefore end on views of the non-centrosomal array that localize very close to the centrosomal array, making it difficult for the reader to distinguish the two. This becomes especially problematic when trying to interpret effects specifically on the non-centrosomal array, e.g. Figure 3D', E'. As the strongest case the authors are using to argue that non-centrosomal microtubules come from the centrosome is that increased g-TuRC expression leads to increased non-centrosomal microtubules, the authors could show xz views of non-centrosomal arrays in these cases as has been done in Figure 6E. Similarly, I find the imaging in new Figure S4I, J incredibly hard to interpret. Can the authors show a movie of the proposed increased comets coming from the centrosome in these conditions rather than a time projection?

Conclusions from g-TuRC localization: The authors stand strong on the myopic view that "de novo nucleation" (defined by the authors I think as the presence of dynamic microtubules, i.e. EB1 dynamics) requires g-TuRC. However, this has been refuted in several systems including the one

referenced by the author:

Line 421: "On the one hand, non-centrosomal microtubules can be de novo nucleated at non-centrosomal MTOCs, and for this process γ -tubulin as part of the γ -TURC is essential 12, 36." Here, the authors cite Feldman & Priess, 2012 because g-TuRC was shown to relocate from the centrosome to the apical non-centrosomal microtubule array. Later work (Sallee et al, 2018) in that system showed g-TuRC to be dispensable to create the apical non-centrosomal microtubule array in intestinal cells, despite the fact that g-TuRC localizes there. This is perhaps a cautionary tale to the authors that the presence (or absence) of g-TuRC cannot necessarily be used to confirm or deny to presence of de novo nucleation. As mentioned in my earlier review, other studies have similar found dynamic microtubule populations in the absence of g-TuRC at both centrosomal and non-centrosomal sites.

Soften language: Similar to the above point, the authors should soften their language throughout the text. They should present the results and speculate about their meaning in the discussion. For example:

"Furthermore, microtubules generated from the remaining active centrosome are released by severing through Katanin and then anchored and stabilised by Patronin, the Drosophila CAMSAP homologue that is quickly recruited to free microtubule minus-ends in placodal cells"

This is speculation based on several pieces of data and it should be presented as such

"This excess amount of microtubules was the consequence of an increased proportion of centrosomes nucleating microtubules in γ -tubulin-EGFP overexpressing embryos (Fig. S4H-J)."

Again, this is speculation based on the data but presented in the main results as fact.

Response to Reviewers' Comments

We are pleased the reviewers found our revised manuscript to have 'effectively addressed' the concerns, 'providing substantial addition', 'adding significant and important experiments that further bolster' our conclusion. We are especially pleased to read that 'the approaches are elegant, the data are clear, and the paper is very well-written'. In the section below we address all remaining concerns by reviewer 2 point by point.

Reviewer #1 (Remarks to the Author):

The authors have effectively addressed my past concerns, providing substantial additions of data addressing mechanisms of molecular interactions, as well as providing broader developmental context and clearer explanation of statistical tests. Overall, the paper should be of broad interest. It dissects major steps of non-centrosomal microtubule array formation in epithelial cells, and shows that this microtubule reorganization promotes actomyosin-based apical constriction and associated tissue morphogenesis in vivo. The approaches are elegant, the data are clear, and the paper is very well-written.

We are very pleased that this reviewer found the experimental additions to have addressed their questions sufficiently, we also feel that the revision has helped strengthen the manuscript and would like to thank for the useful comments.

Reviewer #2 (Remarks to the Author):

Here Gillard et al. present their revised manuscript, adding significant and important experiments that further bolster their claim that non-centrosomal microtubules in epithelial cells of the *Drosophila* salivary gland are captured after release from the centrosome. Release and capture has been proposed for many years as a mechanism to generate non-centrosomal microtubule arrays, however little direct evidence exists, especially in vivo, to support this mechanism. The authors now add additional data, including following Jupiter:GFP as it moves away from centrosomes (Fig. 4E, F) and the ability of Patronin to localize to free microtubule minus ends that have been created by severing. These experiments strengthen the authors' claim, however, I think that it is still open to interpretation whether they have demonstrated release and capture. As they say in their rebuttal, they still present plenty of corroborating evidence, however correlation does not equal causation. Ultimately, it will be up to the reader to decide, but I think the authors could significantly strengthen their paper with the following revisions:

Major Comments:

Improve Imaging: While I feel reasonable confident that the authors have found several conditions to affect the centrosomal microtubule array, I am still left unconvinced by the phenotypes surrounding the non-centrosomal microtubule arrays. The issue arises from the fact that the authors show apical and therefore end on views of the non-centrosomal array that localize very close to the centrosomal array, making it difficult for the reader to distinguish the two. This becomes especially problematic when trying to interpret effects specifically on the non-centrosomal array, e.g. Figure 3D', E'. As the strongest case the authors are using to argue that non-centrosomal microtubules come from the centrosome is that increased g-TuRC expression leads to increased non-centrosomal microtubules, the authors could show xz views of non-centrosomal arrays in these cases as has been done in Figure 6E.

To illustrate the increased level in longitudinal non-centrosomal microtubules that we observe when γ -tubulin is overexpressed, we now provide images that show the whole view of the placodes magnified as panels in Figure 3N, O combined with cross-sectional views. These are now provided in Supplemental Figure S4 F-G'. The cross-section panels show the increase of longitudinal microtubules labelled with tyrosinated α -tubulin across the whole placode when γ -tubulin is overexpressed, whereas in the control these longitudinal microtubules are confined to the area closer to the invagination pit.

Similarly, I find the imaging in new Figure S4I, J incredibly hard to interpret. Can the authors show a movie of the proposed increased comets coming from the centrosome in these conditions rather than a time projection?

*We have changed the panels in Figure S4 to make the change in nucleation clearer and are also including the corresponding movies as Supplemental Movie 2. Imaging EB1 comets in very small apical areas (only a couple of micrometres across) in a bending tissue is challenging, and even more so in this set-up as we have to rely on an EB1 tagged with mCherry (as the *ncd::\gamma*-tubulin-EGFP transgene is already GFP) that is notoriously much fainter and harder to image than the GFP-tagged version of EB1. We hope that the adjusted panels and now Supplemental Movie 2 are clarifying the results sufficiently.*

Conclusions from g-TuRC localization: The authors stand strong on the myopic view that “de novo nucleation” (defined by the authors I think as the presence of dynamic microtubules, i.e. EB1 dynamics) requires g-TuRC. However, this has been refuted in several systems including the one referenced by the author:

Line 421: “On the one hand, non-centrosomal microtubules can be de novo nucleated at non-centrosomal MTOCs, and for this process γ -tubulin as part of the γ -TURC is essential 12, 36.”

Here, the authors cite Feldman & Priess, 2012 because g-TuRC was shown to relocate from the centrosome to the apical non-centrosomal microtubule array. Later work (Sallee et al, 2018) in that system showed g-TuRC to be dispensable to create the apical non-centrosomal microtubule array in intestinal cells, despite the fact that g-TuRC localizes there. This is perhaps a cautionary tale to the authors that the presence (or absence) of g-TuRC cannot necessarily be used to confirm or deny to presence of de novo nucleation. As mentioned in my earlier review, other studies have similar found dynamic microtubule populations in the absence of g-TuRC at both centrosomal and non-centrosomal sites.

*We do not disagree with the reviewer or the Feldman (2018) paper that dynamic microtubule arrays can be maintained without γ -tubulin. What appears to be different in our system compared to the *C.elegans* gut as well as in comparison to *Drosophila* tracheae is that in the salivary gland placode one centrosome per cell retains nucleation capacity (as we clearly show with EB1-GFP comets emanating from these centrosomes), and concomitantly all γ -tubulin is only found at centrosomes. Nonetheless such centrosomally nucleated microtubules lead to a non-centrosomal array.*

We have adjusted our language in the manuscript to present a less ‘myopic’ view.

Soften language: Similar to the above point, the authors should soften their language throughout the text. They should present the results and speculate about their meaning in the discussion. For example:

“Furthermore, microtubules generated from the remaining active centrosome are released by severing through Katanin and then anchored and stabilised by Patronin, the *Drosophila* CAMSAP homologue that is quickly recruited to free microtubule minus-ends in placodal cells”

This is speculation based on several pieces of data and it should be presented as such

“This excess amount of microtubules was the consequence of an increased proportion of centrosomes nucleating microtubules in γ -tubulin-EGFP overexpressing embryos (Fig. S4H-J).”

Again, this is speculation based on the data but presented in the main results as fact.

We have rephrased the description of the results to more cautious statements of possibility throughout the text and have restricted our speculations as suggested to the discussion. All cautious phrasing in the manuscript has been highlighted in the edit-on version.